# DISTRIBUTIONALLY ROBUST RECOURSE ACTION

**Duy Nguyen[1], Ngoc Bui[1], Viet Anh Nguyen[2]**
[1]VinAI Research, Vietnam
[2]The Chinese University of Hong Kong

## ABSTRACT

A recourse action aims to explain a particular algorithmic decision by showing one specific way in which the instance could be modified to receive an alternate outcome. Existing recourse generation methods often assume that the machine learning model does not change over time. However, this assumption does not always hold in practice because of data distribution shifts, and in this case, the recourse action may become invalid. To redress this shortcoming, we propose the Distributionally Robust Recourse Action (DiRRAc) framework, which generates a recourse action that has a high probability of being valid under a mixture of model shifts. We formulate the robustified recourse setup as a min-max optimization problem, where the max problem is specified by Gelbrich distance over an ambiguity set around the distribution of model parameters. Then we suggest a projected gradient descent algorithm to find a robust recourse according to the min-max objective. We show that our DiRRAc framework can be extended to hedge against the misspecification of the mixture weights. Numerical experiments with both synthetic and three real-world datasets demonstrate the benefits of our proposed framework over state-of-the-art recourse methods.

## 1 INTRODUCTION

Post-hoc explanations of machine learning models are useful for understanding and making reliable predictions in consequential domains such as loan approvals, college admission, and healthcare. Recently, recourse has been rising as an attractive tool to diagnose why machine learning models have made a particular decision for a given instance. A recourse action provides a possible modification of the given instance to receive an alternate decision (Ustun et al., 2019). Consider, for example, the case of loan approvals in which a credit application is rejected. Recourse will offer the reasons for rejection by showing what the application package should have been to get approved. A concrete example of a recourse in this case may be "the monthly salary should be higher by $500" or "20% of the current debt should be reduced".

A recourse action has a positive, forward-looking meaning: they list out a directive modification that a person should implement so that they can get a more favorable outcome in the future. If a machine learning system can provide the negative outcomes with the corresponding recourse action, it can improve user engagement and boost the interpretability at the same time (Ustun et al., 2019; Karimi et al., 2021b). Explanations thus play a central role in the future development of human-computer interaction as well as human-centric machine learning.

Despite its attractiveness, providing recourse for the negative instances is not a trivial task. For real-world implementation, designing a recourse needs to strike an intricate balance between conflicting criteria. First and foremost, a recourse action should be feasible: if the prescribed action is taken, then the prediction of a machine learning model should be flipped. Further, to avoid making a drastic change to the characteristics of the input instance, a framework for generating recourse should minimize the cost of implementing the recourse action. An algorithm for finding recourse must make changes to only features that are actionable and should leave immutable features (relatively) unchanged. For example, we must consider the date of birth as an immutable feature; in contrast, we can consider salary or debt amount as actionable features.

Various solutions have been proposed to provide recourses for a model prediction (Karimi et al., 2021b; Stepin et al., 2021; Artelt & Hammer, 2019; Pawelczyk et al., 2021; 2020; Verma et al.,

2020). For instance, Ustun et al. (2019) used an integer programming approach to obtain actionable recourses and also provide a feasibility guarantee for linear models. Karimi et al. (2020) proposed a model-agnostic approach to generate the nearest counterfactual explanations and focus on structured data. Dandl et al. (2020) proposed a method that finds the counterfactual by solving a multi-objective optimization problem. Recently, Russell (2019) and Mothilal et al. (2020) focus on finding a set of multiple diverse recourse actions, where the diversity is imposed by a rule-based approach or by internalizing a determinant point process cost in the objective function.

These aforementioned approaches make a fundamental assumption that the machine learning model does not change over time. However, the dire reality suggests that this assumption rarely holds. In fact, data shifts are so common nowadays in machine learning that they have sparkled the emerging field of domain generalization and domain adaptation. Organizations usually retrain models as a response to data shifts, and this induces corresponding shifts in the machine learning models parameters, which in turn cause serious concerns for the feasibility of the recourse action in the future (Rawal et al., 2021). In fact, all of the aforementioned approaches design the action which is feasible only with the *current* model parameters, and they provide no feasibility guarantee for the *future* parameters. If a recourse action fails to generate a favorable outcome in the future, then the recourse action may become less beneficial (Venkatasubramanian & Alfano, 2020), the pledge of a brighter outcome is shattered, and the trust in the machine learning system is lost (Rudin, 2019).

To tackle this challenge, Upadhyay et al. (2021) proposed ROAR, a framework for generating instance-level recourses that are robust to shifts in the underlying predictive model. ROAR used a robust optimization approach that hedges against an uncertainty set containing plausible values of the future model parameters. However, it is well-known that robust optimization solutions can be overly conservative because they may hedge against a pathological parameter in the uncertainty set (Ben-Tal et al., 2017; Roos & den Hertog, 2020). A promising approach that can promote robustness while at the same time prevent from over-conservatism is the distributionally robust optimization framework (El Ghaoui et al., 2003; Delage & Ye, 2010; Rahimian & Mehrotra, 2019; Bertsimas et al., 2018). This framework models the future model parameters as random variables whose underlying distribution is unknown but is likely to be contained in an ambiguity set. The solution is designed to counter the worst-case distribution in the ambiguity set in a min-max sense. Distributionally robust optimization is also gaining popularity in many estimation and prediction tasks in machine learning (Namkoong & Duchi, 2017; Kuhn et al., 2019).

**Contributions.** This paper combines ideas and techniques from two principal branches of explainable artificial intelligence: counterfactual explanations and robustness to resolve the recourse problem under uncertainty. Concretely, our main contributions are the following:

1. We propose the framework of Distributionally Robust Recourse Action (DiRRAc) for designing a recourse action that is robust to mixture shifts of the model parameters. Our DiRRAc maximizes the probability that the action is feasible with respect to a mixture shift of model parameters while at the same time confines the action in the neighborhood of the input instance. Moreover, the DiRRAc model also hedges against the misspecification of the nominal distribution using a min-max form with a mixture ambiguity set prescribed by moment information.

2. We reformulate the DiRRAc problem into a finite-dimensional optimization problem with an explicit objective function. We also provide a projected gradient descent to solve the problem.

3. We extend our DiRRAc framework along several axis to handle mixture weight uncertainty, to minimize the worst-case component probability of receiving the unfavorable outcome, and also to incorporate the Gaussian parametric information.

We first describe the recourse action problem with mixture shifts in Section 2. In Section 3, we present our proposed DiRRAc framework, its reformulation and the numerical routine for solving it. The extension to the parametric Gaussian setting will be discussed in Section 4. Section 5 reports the numerical experiments showing the benefits of the DiRRAc framework and its extensions.

**Notations.** For each integer $K$, we have $[K] = \{1, \ldots, K\}$. We use $\mathbb{S}_+^d$ ($\mathbb{S}_{++}^d$) to denote the space of symmetric positive semidefinite (definite) matrices. For any $A \in \mathbb{R}^{m \times m}$, the trace operator is $\mathrm{Tr}\left[A\right] = \sum_{i=1}^d A_{ii}$. If a distribution $\mathbb{Q}_k$ has mean $\mu_k$ and covariance matrix $\Sigma_k$, we write $\mathbb{Q}_k \sim (\mu_k, \Sigma_k)$. If additionally $\mathbb{Q}_k$ is Gaussian, we write $\mathbb{Q}_k \sim \mathcal{N}(\mu_k, \Sigma_k)$. Writing $\mathbb{Q} \sim (\mathbb{Q}_k, p_k)_{k \in [K]}$ means $\mathbb{Q}$ is a mixture of $K$ components, the $k$-th component has weight $p_k$ and distribution $\mathbb{Q}_k$.

## 2 RECOURSE ACTION UNDER MIXTURE SHIFTS

We consider a binary classification setting with label $\mathcal{Y} = \{0, 1\}$, where $0$ represents the unfavorable outcome while $1$ denotes the favorable one. The covariate space is $\mathbb{R}^d$, and any linear classifier $\mathcal{C}_\theta : \mathbb{R}^d \to \mathcal{Y}$ characterized by the $d$-dimensional parameter $\theta$ is of the form

$$\mathcal{C}_\theta(x) = \begin{cases} 1 & \text{if } \theta^\top x \geq 0, \\ 0 & \text{otherwise.} \end{cases}$$

Note that the bias term can be internalized into $\theta$ by adding an extra dimension, and thus it is omitted.

Suppose that at this moment ($t = 0$), the current classifier is parametrized by $\theta_0$, and we are given an input instance $x_0 \in \mathbb{R}^d$ with *un*favorable outcome, that is, $\mathcal{C}_{\theta_0}(x_0) = 0$. One period of time from now ($t = 1$), the parameters of the predictive model will change stochastically and are represented by a $d$-dimensional random vector $\tilde{\theta}$. This paper focuses on finding a recourse action $x$ which is reasonably close to the instance $x_0$, and at the same time, has a high probability of receiving a favorable outcome in the future. Figure 1 gives a bird's eye view of the setup.

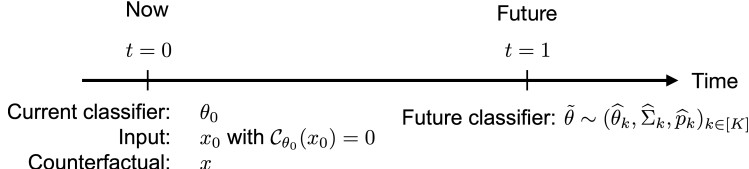

Figure 1: A canonical setup of the recourse action under mixture shifts problem.

To measure the closeness between the action $x$ and the input $x_0$, we assume that the covariate space is endowed with a non-negative, continuous cost function $c$. In addition, suppose temporarily that $\tilde{\theta}$ follows a distribution $\widehat{\mathbb{P}}$. Because maximizing the probability of the favorable outcome is equivalent to minimizing the probability of the unfavorable outcome, the recourse can be found by solving

$$\min \left\{ \widehat{\mathbb{P}}(\mathcal{C}_{\tilde{\theta}}(x) = 0) \; : \; x \in \mathbb{X}, \; c(x, x_0) \leq \delta \right\}. \tag{1}$$

The parameter $\delta \geq 0$ in (1) governs how far a recourse action can be from the input instance $x_0$. Note that we constrain $x$ in a set $\mathbb{X}$ which captures operational constraints, for example, the highest education of a credit applicant should not be decreasing over time.

In this paper, we model the random vector $\tilde{\theta}$ using a finite mixture of distributions with $K$ components, the mixture weights are $\widehat{p}$ satisfying $\sum_{k \in [K]} \widehat{p}_k = 1$. Each component in the mixture represents one specific type of model shifts: the weights $\widehat{p}$ reflect the proportion of the shift types while the component distribution $\widehat{\mathbb{P}}_k$ represents the (conditional) distribution of the future model parameters in the $k$-th shift. Further information on mixture distributions and their applications in machine learning can be found in (Murphy, 2012, §3.5). Note that the mixture model is not a strong assumption. It is well-known that the Gaussian mixture model is a universal approximator of densities, in the sense that any smooth density can be approximated with any specific nonzero amount of error by a Gaussian mixture model with enough components (Goodfellow et al., 2016; McLachlan & Peel, 2000). Thus, our mixture models are flexible enough to hedge against distributional perturbations of the parameters under large values of $K$. The design of the ambiguity set to handle ambiguous mixture weights and under the Gaussian assumption is extensively studied in the literature on distributionally robust optimization (Hanasusanto et al., 2015; Chen & Xie, 2021).

If each $\widehat{\mathbb{P}}_k$ is a Gaussian distribution $\mathcal{N}(\widehat{\theta}_k, \widehat{\Sigma}_k)$, then $\widehat{\mathbb{P}}$ is a mixture of Gaussian distributions. The objective of problem (1) can be expressed as

$$\widehat{\mathbb{P}}(\mathcal{C}_{\tilde{\theta}}(x) = 0) = \sum_{k \in [K]} \widehat{p}_k \widehat{\mathbb{P}}_k(\mathcal{C}_{\tilde{\theta}}(x) = 0) = \sum_{k \in [K]} \widehat{p}_k \Phi\left(\frac{-x^\top \widehat{\theta}_k}{\sqrt{x^\top \widehat{\Sigma}_k x}}\right),$$

where the first equality follows from the law of conditional probability, and $\Phi$ is the cumulative distribution function of a standard Gaussian distribution. Under the Gaussian assumption, we can solve (1) using a projected gradient descent type of algorithm (Boyd & Vandenberghe, 2004).

**Remark 2.1** (Nonlinear models). *Our analysis focuses on linear classifiers, which is a common setup in the literature (Upadhyay et al., 2021; Ustun et al., 2019; Rawal et al., 2021; Karimi et al., 2020; Wachter et al., 2018; Ribeiro et al., 2016). To extend to nonlinear classifiers, we can follow a similar approach as in Rawal & Lakkaraju (2020b) and Upadhyay et al. (2021) by first using LIME Ribeiro et al. (2016) to approximate the nonlinear classifiers locally with an interpretable linear model, then subsequently applying our framework.*

## 3 DISTRIBUTIONALLY ROBUST RECOURSE ACTION FRAMEWORK

Our Distributionally Robust Recourse Action (DiRRAc) framework robustifies formulation (1) by relaxing the parametric assumption and hedging against distribution misspecification. First, we assume that the mixture components $\widehat{\mathbb{P}}_k$ are specified only through moment information, and no particular parametric form of the distribution is imposed. In effect, $\widehat{\mathbb{P}}_k$ is assumed to have mean vector $\widehat{\theta}_k \in \mathbb{R}^d$ and positive definite covariance matrix $\widehat{\Sigma}_k \succ 0$. Second, we leverage ideas from distributionally robust optimization to propose a min-max formulation of (1), in which we consider an *ambiguity set* which contains a family of probability distributions that are sufficiently close to the nominal distribution $\widehat{\mathbb{P}}$. We prescribe the ambiguity set using Gelbrich distance (Gelbrich, 1990).

**Definition 3.1** (Gelbrich distance). *The Gelbrich distance $\mathbb{G}$ between two tuples $(\theta, \Sigma) \in \mathbb{R}^d \times \mathbb{S}_+^d$ and $(\widehat{\theta}, \widehat{\Sigma}) \in \mathbb{R}^d \times \mathbb{S}_+^d$ amounts to $\mathbb{G}((\theta, \Sigma), (\widehat{\theta}, \widehat{\Sigma})) \triangleq \sqrt{\|\theta - \widehat{\theta}\|_2^2 + \mathrm{Tr}\left[\Sigma + \widehat{\Sigma} - 2(\widehat{\Sigma}^{\frac{1}{2}} \Sigma \widehat{\Sigma}^{\frac{1}{2}})^{\frac{1}{2}}\right]}$.*

It is easy to verify that $\mathbb{G}$ is non-negative, symmetric and it vanishes to zero if and only if $(\theta, \Sigma) = (\widehat{\theta}, \widehat{\Sigma})$. Further, $\mathbb{G}$ is a distance on $\mathbb{R}^d \times \mathbb{S}_+^d$ because it coincides with the type-2 Wasserstein distance between two Gaussian distributions $\mathcal{N}(\mu, \Sigma)$ and $\mathcal{N}(\widehat{\mu}, \widehat{\Sigma})$ (Givens & Shortt, 1984). Distributionally robust formulations with moment information prescribed by the $\mathbb{G}$ distance are computationally tractable under mild conditions, deliver reasonable performance guarantees and also generate a conservative approximation of the Wasserstein distributionally robust optimization problem (Kuhn et al., 2019; Nguyen et al., 2021).

In this paper, we use the Gelbrich distance $\mathbb{G}$ to form a neighborhood around each $\widehat{\mathbb{P}}_k$ as

$$\mathcal{B}_k(\widehat{\mathbb{P}}_k) \triangleq \left\{ \mathbb{Q}_k : \mathbb{Q}_k \sim (\theta_k, \Sigma_k), \ \mathbb{G}((\theta_k, \Sigma_k), (\widehat{\theta}_k, \widehat{\Sigma}_k)) \leq \rho_k \right\}.$$

Intuitively, one can view $\mathcal{B}_k(\widehat{\mathbb{P}}_k)$ as a ball centered at the nominal component $\widehat{\mathbb{P}}_k$ of radius $\rho_k \geq 0$ prescribed using the distance $\mathbb{G}$. This component set $\mathcal{B}_k(\widehat{\mathbb{P}}_k)$ is non-parametric, and the first two moments of $\mathbb{Q}_k$ are sufficient to decide whether $\mathbb{Q}_k$ belongs to $\mathcal{B}_k(\widehat{\mathbb{P}}_k)$. Moreover, if $\mathbb{Q}_k \in \mathcal{B}_k(\widehat{\mathbb{P}}_k)$, then any distribution $\mathbb{Q}_k'$ with the same mean vector and covariance matrix as $\mathbb{Q}_k$ also belongs to $\mathcal{B}_k(\widehat{\mathbb{P}}_k)$. Notice that even when the radius $\rho_k$ is zero, the component set $\mathcal{B}_k(\widehat{\mathbb{P}}_k)$ does not collapse into a singleton. Instead, if $\rho_k = 0$ then $\mathcal{B}_k(\widehat{\mathbb{P}}_k)$ still contains *all* distributions of the same moment $(\widehat{\theta}_k, \widehat{\Sigma}_k)$ with the nominal component distribution $\widehat{\mathbb{P}}_k$, and consequentially it possesses the robustification effects against the parametric assumption on $\widehat{\mathbb{P}}_k$. The component sets are utilized to construct the ambiguity set for the mixture distribution as

$$\mathcal{B}(\widehat{\mathbb{P}}) \triangleq \left\{ \mathbb{Q} : \ \exists \mathbb{Q}_k \in \mathcal{B}_k(\widehat{\mathbb{P}}_k) \ \forall k \in [K] \text{ such that } \mathbb{Q} \sim (\mathbb{Q}_k, \widehat{p}_k)_{k \in [K]} \right\}.$$

Any $\mathbb{Q} \in \mathcal{B}(\widehat{\mathbb{P}})$ is also a mixture distribution with $K$ components, with the same mixture weights $\widehat{p}$. Thus, $\mathcal{B}(\widehat{\mathbb{P}})$ contains all perturbations of $\widehat{\mathbb{P}}$ induced separately on each component by $\mathcal{B}_k(\widehat{\mathbb{P}}_k)$.

We are now ready to introduce our DiRRAc model, which is a min-max problem of the form

$$\begin{aligned}
\inf_{x \in \mathbb{X}} \quad & \sup_{\mathbb{Q} \in \mathcal{B}(\widehat{\mathbb{P}})} \mathbb{Q}(\mathcal{C}_{\tilde{\theta}}(x) = 0) \\
\text{s.t.} \quad & c(x, x_0) \leq \delta \\
& \sup_{\mathbb{Q}_k \in \mathcal{B}_k(\widehat{\mathbb{P}}_k)} \mathbb{Q}_k(\mathcal{C}_{\tilde{\theta}}(x) = 0) < 1 \qquad \forall k \in [K].
\end{aligned} \tag{2}$$

The objective of (2) is to minimize the worst-case probability of unfavorable outcome of the recourse action. Moreover, the last constraint imposes that for each component, the worst-case conditional

probability of unfavorable outcome should be strictly less than one. Put differently, this last constraint requires that the action should be able to lead to favorable outcome for *any* distribution in $\mathcal{B}_k(\widehat{\mathbb{P}}_k)$. By definition, each supremum subproblem in (2) is an infinite-dimensional maximization problem over the space of probability distributions, and thus it is inherently difficult. Fortunately, because we use the Gelbrich distance to prescribe the set $\mathcal{B}_k(\widehat{\mathbb{P}}_k)$, we can solve these maximization problems analytically. This consequentially leads to a closed-form reformulation of the DiRRAc model into a finite-dimensional problem. Next, we will reformulate the DiRRAc problem (2), provide a sketch of the proof and propose a numerical solution routine.

## 3.1 REFORMULATION OF DIRRAC

Each supremum in (2) is an infinite-dimensional optimization problem on the space of probability distributions. We now show that (2) can be reformulated as a finite-dimensional problem. Towards this end, let $\mathcal{X}$ be the following $d$-dimensional set.

$$\mathcal{X} \triangleq \left\{ x \in \mathbb{X} : \; c(x, x_0) \leq \delta, \quad -\widehat{\theta}_k^\top x + \rho_k \|x\|_2 < 0 \quad \forall k \in [K] \; \right\}. \tag{3}$$

The next theorem asserts that the DiRRAc problem (2) can be reformulated as a $d$-dimensional optimization problem with an explicit, but complicated, objective function.

**Theorem 3.2** (Equivalent form of DiRRAc). *Problem (2) is equivalent to the finite-dimensional optimization problem*

$$\inf_{x \in \mathcal{X}} \sum_{k \in [K]} \widehat{p}_k f_k(x)^2, \tag{4}$$

*where the function $f_k$ admits the closed-form expression*

$$f_k(x) = \frac{\rho_k \widehat{\theta}_k^\top x \|x\|_2 + \sqrt{x^\top \widehat{\Sigma}_k x} \sqrt{(\widehat{\theta}_k^\top x)^2 + x^\top \widehat{\Sigma}_k x - \rho_k^2 \|x\|_2^2}}{(\widehat{\theta}_k^\top x)^2 + x^\top \widehat{\Sigma}_k x}.$$

Next, we sketch a proof of Theorem 3.2 and a solution procedure to solve problem (4).

## 3.2 PROOF SKETCH

For any component $k \in [K]$, define the following worst-case probability of unfavorable outcome

$$f_k(x) \triangleq \sup_{\mathbb{Q}_k \in \mathcal{B}_k(\widehat{\mathbb{P}}_k)} \mathbb{Q}_k(\mathcal{C}_{\tilde{\theta}}(x) = 0) = \sup_{\mathbb{Q}_k \in \mathcal{B}_k(\widehat{\mathbb{P}}_k)} \mathbb{Q}_k(\tilde{\theta}^\top x \leq 0) \qquad \forall k \in [K]. \tag{5}$$

To proceed, we rely on the following elementary result from (Nguyen, 2019, Lemma 3.31).

**Lemma 3.3** (Worst-case Value-at-Risk). *For any $x \in \mathbb{R}^d$ and $\beta \in (0, 1)$, we have*

$$\inf \left\{ \tau : \sup_{\mathbb{Q}_k \in \mathcal{B}_k(\widehat{\mathbb{P}}_k)} \mathbb{Q}_k(\tilde{\theta}^\top x \leq -\tau) \leq \beta \right\} = -\widehat{\theta}_k^\top x + \sqrt{\frac{1-\beta}{\beta}} \sqrt{x^\top \widehat{\Sigma}_k x} + \frac{\rho_k}{\sqrt{\beta}} \|x\|_2. \tag{6}$$

Note that the left-hand side of (6) is the worst-case Value-at-Risk with respect to the ambiguity set $\mathcal{B}_k(\widehat{\mathbb{P}}_k)$. Leveraging this result, the next proposition provides the analytical form of $f_k(x)$.

**Proposition 3.4** (Worst-case probability). *For any $k \in [K]$ and $(\widehat{\theta}_k, \widehat{\Sigma}_k, \rho_k) \in \mathbb{R}^d \times \mathbb{S}_+^d \times \mathbb{R}_+$, define the following constants $A_k \triangleq -\widehat{\theta}_k^\top x$, $B_k \triangleq \sqrt{x^\top \widehat{\Sigma}_k x}$, and $C_k \triangleq \rho_k \|x\|_2$. We have*

$$f_k(x) \triangleq \sup_{\mathbb{Q}_k \in \mathcal{B}_k(\widehat{\mathbb{P}}_k)} \mathbb{Q}_k(\tilde{\theta}^\top x \leq 0) = \begin{cases} 1 & \text{if } A_k + C_k \geq 0, \\ \left( \frac{-A_k C_k + B_k \sqrt{A_k^2 + B_k^2 - C_k^2}}{A_k^2 + B_k^2} \right)^2 \in (0, 1) & \text{if } A_k + C_k < 0. \end{cases}$$

The proof of Theorem 3.2 follows by noticing that the DiRRAc problem (2) can be reformulated using the elementary functions $f_k$ as

$$\min_{x \in \mathbb{X}} \left\{ \sum_{k \in [K]} \widehat{p}_k f_k(x) \; : \; c(x, x_0) \leq \delta, \; f_k(x) < 1 \quad \forall k \in [K] \right\},$$

where the objective function follows from the definition of the set $\mathcal{B}(\widehat{\mathbb{P}})$. It suffices now to combine with Proposition 3.4 to obtain the necessary result. The detailed proof is relegated to the Appendix. Next we propose a projected gradient descent algorithm to solve the problem (4).

### 3.3 PROJECTED GRADIENT DESCENT ALGORITHM

We consider in this section an iterative numerical routine to solve the DiRRAc problem in the equivalent form (4). First, notice that the second constraint that defines $\mathcal{X}$ in (3) is a strict inequality, thus the set $\mathcal{X}$ is open. We thus modify slightly this constraint by considering the following set

$$\mathcal{X}_{\varepsilon} = \left\{ x \in \mathbb{X} \; : \; c(x, x_0) \le \delta, \; -\widehat{\theta}_k^\top x + \rho_k \|x\|_2 \le -\varepsilon \quad \forall k \in [K] \right\}$$

for some value $\varepsilon > 0$ sufficiently small. Moreover, if the parameter $\delta$ is too small, it may happen that the set $\mathcal{X}_\varepsilon$ becomes empty. Define $\delta_{\min} \in \mathbb{R}_+$ as the optimal value of the following problem

$$\inf \left\{ c(x, x_0) \; : \; x \in \mathbb{X}, \; -\widehat{\theta}_k^\top x + \rho_k \|x\|_2 \le -\varepsilon \quad \forall k \in [K] \right\}. \tag{7}$$

Then it is easy to see that $\mathcal{X}_\varepsilon$ is non-empty whenever $\delta \ge \delta_{\min}$. In addition, because $c$ is continuous and $\mathbb{X}$ is closed, the set $\mathcal{X}_\varepsilon$ is compact. In this case, we can consider problem (4) with the feasible set being $\mathcal{X}_\varepsilon$, for which the optimal solution is guaranteed to exist. Let us now define the projection operator $\mathrm{Proj}_{\mathcal{X}_\varepsilon}$ as $\mathrm{Proj}_{\mathcal{X}_\varepsilon}(x') \triangleq \arg\min \left\{ \|x - x'\|_2^2 \; : \; x \in \mathcal{X}_\varepsilon \right\}$. If $\mathbb{X}$ is convex and $c(\cdot, x_0)$ is a convex function, then $\mathcal{X}_\varepsilon$ is also convex, and the projection operation can be efficiently computed using convex optimization.

In particular, suppose that $c(x, x_0) = \|x - x_0\|_2$ is the Euclidean norm and $\mathbb{X}$ is second-order cone representable, then the projection is equivalent to a second-order cone program, and can be solved using off-the-shelf solvers such as GUROBI Gurobi Optimization, LLC (2021) or Mosek (MOSEK ApS, 2019). The projection operator $\mathrm{Proj}_{\mathcal{X}_\varepsilon}$ now forms the building block of a projected gradient descent algorithm with a backtracking linesearch. The details regarding the algorithm, along with the convergence guarantee, are presented in Appendix E.

To conclude this section, we visualize the geometrical intuition of our method in Figure 2.

Figure 2: The feasible set $\mathcal{X}$ in (3) is shaded in blue. The circular arc represents the proximity boundary $c(x, x_0) = \delta$ with $c$ being an Euclidean distance. Dashed lines represent the hyperplane $-\widehat{\theta}_k^\top x = 0$ for different $k$, while elliptic curves represent the robust margin $-\widehat{\theta}_k^\top x + \rho_k \|x\| = 0$ with matching color. Increasing the ambiguity size $\rho_k$ brings the elliptic curves towards the top-right corner and farther away from the dash lines. The set $\mathcal{X}$ taken as the intersection of elliptical and promixity constraints will move deeper into the interior of the favorable prediction region, resulting in more robust recourses.

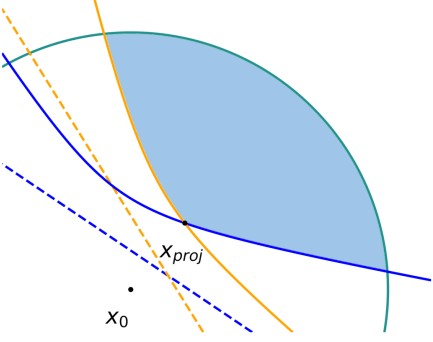

### 4 GAUSSIAN DIRRAC FRAMEWORK

We here revisit the Gaussian assumption on the component distributions, and propose the parametric Gaussian DiRRAc framework. We make the temporary assumption that $\widehat{\mathbb{P}}_k$ are Gaussian for all $k \in [K]$, and we will robustify against only the misspecification of the nominal mean vector and covariance matrix $(\widehat{\theta}_k, \widehat{\Sigma}_k)$. To do this, we first construct the Gaussian component ambiguity sets

$$\forall k : \quad \mathcal{B}_k^{\mathcal{N}}(\widehat{\mathbb{P}}_k) \triangleq \left\{ \mathbb{Q}_k : \; \mathbb{Q}_k \sim \mathcal{N}(\theta_k, \Sigma_k), \; \mathbb{G}((\theta_k, \Sigma_k), (\widehat{\theta}_k, \widehat{\Sigma}_k)) \le \rho_k \right\},$$

where the superscript emphasizes that the ambiguity sets are neighborhoods in the space of Gaussian distributions. The resulting ambiguity set for the mixture distribution is

$$\mathcal{B}^{\mathcal{N}}(\widehat{\mathbb{P}}) = \left\{ \mathbb{Q} : \; \exists \mathbb{Q}_k \in \mathcal{B}_k^{\mathcal{N}}(\widehat{\mathbb{P}}_k) \; \forall k \in [K] \text{ such that } \mathbb{Q} \sim (\mathbb{Q}_k, \widehat{p}_k)_{k \in [K]} \right\}.$$

The Gaussian DiRRAc problem is formally defined as

$$
\begin{aligned}
\min_{x \in \mathbb{X}} \quad & \sup_{\mathbb{Q} \in \mathcal{B}^{\mathcal{N}}(\widehat{\mathbb{P}})} \mathbb{Q}(\mathcal{C}_{\tilde{\theta}}(x) = 0) \\
\text{s.t.} \quad & c(x, x_0) \leq \delta \\
& \sup_{\mathbb{Q}_k \in \mathcal{B}_k^{\mathcal{N}}(\widehat{\mathbb{P}}_k)} \mathbb{Q}_k(\mathcal{C}_{\tilde{\theta}}(x) = 0) < \tfrac{1}{2} \qquad \forall k \in [K].
\end{aligned}
\tag{8}
$$

Similar to Section 3, we will provide the reformulation of the Gaussian DiRRAc formulation and a sketch of the proof in the sequence. Note that the last constraint in (8) has margin $\frac{1}{2}$ instead of 1 as in the DiRRAc problem (2). The detailed reason will be revealed in the proof sketch in Section 4.2.

## 4.1 REFORMULATION OF GAUSSIAN DiRRAc

Remind that the feasible set $\mathcal{X}$ is defined as in (3). The next theorem asserts the equivalent form of the Gaussian DiRRAc problem (8).

**Theorem 4.1** (Gaussian DiRRAc reformulation). *The Gaussian DiRRAc problem (8) is equivalent to the finite-dimensional optimization problem*

$$
\min_{x \in \mathcal{X}} 1 - \sum_{k \in [K]} \widehat{p}_k \Phi(g_k(x)),
\tag{9}
$$

*where the function $g_k$ admits the closed-form expression*

$$
g_k(x) = \frac{(\widehat{\theta}_k^\top x)^2 - \rho_k^2 \|x\|_2^2}{\widehat{\theta}_k^\top x \sqrt{x^\top \widehat{\Sigma}_k x} + \rho_k \|x\|_2 \sqrt{(\widehat{\theta}_k^\top x)^2 + x^\top \widehat{\Sigma}_k x - \rho_k^2 \|x\|_2^2}}.
$$

Problem (9) can be solved using the projected gradient descent algorithm discussed in Section 3.3.

## 4.2 PROOF SKETCH

The proof of Theorem 4.1 relies on the following analytical form of the worst-case Value-at-Risk (VaR) under parametric Gaussian ambiguity set (Nguyen, 2019, Lemma 3.31).

**Lemma 4.2** (Worst-case Gaussian VaR). *For any $x \in \mathbb{R}^d$ and $\beta \in (0, \frac{1}{2}]$, let $t = \Phi^{-1}(1 - \beta)$. Then*

$$
\inf \left\{ \tau : \sup_{\mathbb{Q}_k \in \mathcal{B}_k^{\mathcal{N}}(\widehat{\mathbb{P}}_k)} \mathbb{Q}_k(\tilde{\theta}^\top x \leq -\tau) \leq \beta \right\} = -\widehat{\theta}_k^\top x + t \sqrt{x^\top \widehat{\Sigma}_k x} + \rho \sqrt{1 + t^2} \|x\|_2.
\tag{10}
$$

It is important to note that Lemma 4.2 is only valid for $\beta \in (0, 0.5]$. Indeed, for $\beta > \frac{1}{2}$, evaluating the infimum problem in the left-hand side of (10) requires solving a *non-convex* optimization problem as $t = \Phi^{-1}(1 - \beta) < 0$. As a consequence, the last constraint of the Gaussian DiRRAc formulation (8) is capped at a probability value of 0.5 to ensure the convexity of the feasible set in the reformulation (9). The proof of Theorem 4.1 follows a similar line of argument as for the DiRRAc formulation, with $g_k$ being the worst-case Gaussian probability

$$
g_k(x) \triangleq \sup_{\mathbb{Q}_k \in \mathcal{B}_k^{\mathcal{N}}(\widehat{\mathbb{P}}_k)} \mathbb{Q}_k(\mathcal{C}_{\tilde{\theta}}(x) = 0) = \sup_{\mathbb{Q}_k \in \mathcal{B}_k^{\mathcal{N}}(\widehat{\mathbb{P}}_k)} \mathbb{Q}_k(\tilde{\theta}^\top x \leq 0) \qquad \forall k \in [K].
$$

To conclude this section, we provide a quick sanity check: by setting $K = 1$ and $\rho_1 = 0$, we have a special case in which $\tilde{\theta}$ follows a Gaussian distribution $\mathcal{N}(\widehat{\mu}_1, \widehat{\Sigma}_1)$. Thus, $\tilde{\theta}^\top x \sim \mathcal{N}(\widehat{\mu}_1^\top x, x^\top \widehat{\Sigma}_1 x)$ and it is easy to verify from the formula of $g_1$ in the statement of Theorem 4.1 that $g_1(x) = (\widehat{\theta}_1^\top x)/(x^\top \widehat{\Sigma}_1 x)^{\frac{1}{2}}$, which recovers the value of $\Pr(\tilde{\theta}^\top x \leq 0)$ under the Gaussian distribution.

## 5 NUMERICAL EXPERIMENTS

We compare extensively the performance of our DiRRAc model (2) and Gaussian DiRRAc model (8) against four strong baselines: ROAR (Upadhyay et al., 2021), CEPM (Pawelczyk et al.,

2020), AR (Ustun et al., 2019) and Wachter (Wachter et al., 2018). We conduct the experiments on three real-world datasets (German, SBA, Student). Appendix A provides further comparisons with more baselines: Nguyen et al. (2022), Karimi et al. (2021a) and ensemble variants of ROAR, along with the sensitivity analysis of hyperparameters. Appendix A also contains the details about the datasets and the experimental setup.

**Metrics.** For all experiments, we use the $l_1$ distance $c(x, x_0) = \|x - x_0\|_1$ as the *cost function*. Each dataset contains two sets of data (the present and shifted data). The present data is to train the current classifier for which recourses are generated while the remaining data is used to measure the validity of the generated recourses under model shifts. We choose 20% of the shifted data randomly 100 times and train 100 classifiers respectively. The validity of a recourse is computed as the fraction of the classifiers for which the recourse is valid. We then report the average of the validity of all generated recourses and refer this value as $M_2$ *validity*. We also report $M_1$ *validity*, which is the fraction of the instances for which the recourse is valid with respect to the original classifier.

**Results on real-world data.** We use three real-world datasets which capture different data distribution shifts (Dua & Graff, 2017): (i) the German credit dataset, which captures a correction shift. (ii) the Small Business Administration (SBA) dataset, which captures a temporal shift. (iii) the Student performance dataset, which captures a geospatial shift. Each dataset contains original data and shifted data. We normalize all continuous features to $[0, 1]$. Similar to Mothilal et al. (2020), we use one-hot encodings for categorial features, then consider them as continuous features in $[0, 1]$. To ease the comparison, we choose $K = 1$. The choices of $K$ are discussed further in Appendix A.

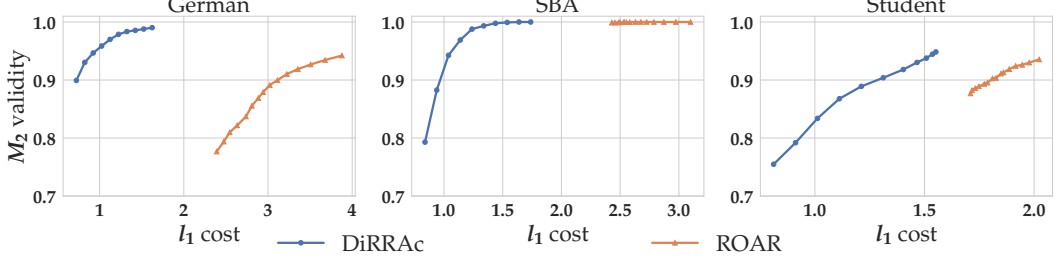

Figure 3: Comparison of $M_2$ validity as a function of the $l_1$ distance between input instance and the recourse for our DiRRAc method and ROAR on real datasets.

Table 1: Benchmark of $M_1$ and $M_2$ validity, $l_1$ and $l_2$ cost for linear models on real datasets.

| Dataset | Methods | $M_1$ validity | $M_2$ validity | $l_1$ cost | $l_2$ cost |
|---|---|---|---|---|---|
| German | AR | **1.00** ± 0.00 | 0.76 ± 0.26 | **0.61** ± 0.40 | 0.43 ± 0.25 |
| | Wachter | **1.00** ± 0.00 | 0.82 ± 0.24 | 0.81 ± 0.51 | **0.41** ± 0.25 |
| | CEPM | **1.00** ± 0.00 | 0.83 ± 0.38 | 1.30 ± 0.02 | 1.02 ± 0.04 |
| | ROAR | **1.00** ± 0.00 | 0.94 ± 0.15 | 3.88 ± 0.54 | 1.61 ± 0.22 |
| | DiRRAc | **1.00** ± 0.00 | **0.99** ± 0.06 | 1.62 ± 0.30 | 1.25 ± 0.21 |
| | Gaussian DiRRAc | **1.00** ± 0.00 | **0.99** ± 0.06 | 1.62 ± 0.30 | 1.05 ± 0.23 |
| SBA | AR | **1.00** ± 0.00 | 0.41 ± 0.18 | **0.61** ± 0.42 | **0.56** ± 0.36 |
| | Wachter | **1.00** ± 0.00 | 0.55 ± 0.22 | 2.30 ± 2.39 | 0.77 ± 0.66 |
| | CEPM | **1.00** ± 0.00 | 0.94 ± 0.24 | 5.30 ± 0.01 | 2.18 ± 0.02 |
| | ROAR | **1.00** ± 0.00 | **1.00** ± 0.00 | 3.10 ± 0.72 | 1.35 ± 0.30 |
| | DiRRAc | **1.00** ± 0.00 | **1.00** ± 0.00 | 1.74 ± 0.44 | 1.34 ± 0.40 |
| | Gaussian DiRRAc | **1.00** ± 0.00 | 0.99 ± 0.02 | 1.60 ± 0.62 | 0.98 ± 0.42 |
| Student | AR | **1.00** ± 0.00 | 0.48 ± 0.19 | **0.29** ± 0.21 | **0.26** ± 0.18 |
| | Wachter | **1.00** ± 0.00 | 0.53 ± 0.19 | 0.60 ± 0.43 | 0.30 ± 0.22 |
| | CEPM | **1.00** ± 0.00 | 0.91 ± 0.15 | 4.52 ± 0.01 | 2.03 ± 0.01 |
| | ROAR | **1.00** ± 0.00 | 0.94 ± 0.10 | 2.02 ± 0.38 | 0.96 ± 0.18 |
| | DiRRAc | **1.00** ± 0.00 | **0.95** ± 0.09 | 1.55 ± 0.34 | 1.07 ± 0.23 |
| | Gaussian DiRRAc | **1.00** ± 0.00 | 0.74 ± 0.18 | 0.78 ± 0.30 | 0.54 ± 0.21 |

We split 80% of the original dataset and train a logistic classifier. This process is repeated 100 times independently to obtain 100 observations of the model parameters. Then we compute the empirical mean and covariance matrix for $(\widehat{\theta}_1, \widehat{\Sigma}_1)$. To evaluate the trade-off between $l_1$ cost and $M_2$ validity of DiRRAc and ROAR, we compute $l_1$ cost and the $M_2$ validity by running DiRRAc with varying

values of $\delta_{\text{add}}$ and ROAR with varying values of $\lambda$. We define $\delta = \delta_{\text{min}} + \delta_{\text{add}}$, $\delta_{\text{min}}$ is specified in (7). Figure 3 shows that the frontiers of DiRRAc dominate the frontiers of ROAR. This indicates that DiRRAc achieves a far smaller $l_1$ cost for the robust recourses than ROAR. Next, we evaluate the $l_1$ and $l_2$ cost, $M_1$ and $M_2$ validity of DiRRAc, ROAR and other baselines. The results in Table 1 demonstrate that DiRRAc has high validity in all three datasets while preserving low costs ($l_1$ and $l_2$ cost) in comparison to ROAR. Our DiRRAc framework consistently outperforms the AR, Wachter, and CEPM in terms of $M_2$ validity.

**Nonlinear models.** Following the previous work as in Rawal et al. (2021); Upadhyay et al. (2021) and Bui et al. (2022), we adapt our DiRRAc framework and other baselines (AR and ROAR) to non-linear models by first generating local linear approximations using LIME (Ribeiro et al., 2016). For each instance $x_0$, we first generate a local linear model for the MLPs classifier 10 times using LIME, each time using 1000 perturbed samples. To estimate $(\widehat{\theta}_1, \widehat{\Sigma}_1)$, we compute the mean and covariance matrix of parameters $\theta_{x_0}$ of 10 local linear models. We randomly choose 10% of the shifted dataset and concatenate with training data of the original dataset 10 times, then train a shifted MLPs classifier. According to Table 2. On the German Credit and Student dataset, DiRRAc has a higher $M_2$ validity than other baselines, and a slightly lower $M_2$ validity on the SBA dataset than ROAR, while maintaining a low $l_1$ cost relative to ROAR and CEPM.

Table 2: Benchmark of $M_1$ and $M_2$ validity, $l_1$ and $l_2$ cost for non-linear models on real datasets.

| Dataset | Methods | $M_1$ validity | $M_2$ validity | $l_1$ cost | $l_2$ cost |
|---------|---------|----------------|----------------|------------|------------|
| German | LIME-AR | $0.72 \pm 0.45$ | $0.71 \pm 0.27$ | $1.05 \pm 0.20$ | $1.00 \pm 0.03$ |
| | Wachter | $\mathbf{1.00} \pm 0.00$ | $0.55 \pm 0.42$ | $\mathbf{0.20} \pm 0.26$ | $\mathbf{0.11} \pm 0.16$ |
| | CEPM | $\mathbf{1.00} \pm 0.00$ | $0.74 \pm 0.40$ | $1.30 \pm 0.01$ | $1.02 \pm 0.00$ |
| | LIME-ROAR | $0.60 \pm 0.49$ | $0.69 \pm 0.27$ | $2.52 \pm 0.20$ | $1.25 \pm 0.07$ |
| | LIME-DiRRAc | $0.78 \pm 0.42$ | $\mathbf{0.75} \pm 0.27$ | $1.14 \pm 0.27$ | $1.02 \pm 0.05$ |
| | LIME-Gaussian DiRRAc | $0.70 \pm 0.46$ | $0.70 \pm 0.31$ | $1.11 \pm 0.26$ | $1.00 \pm 0.06$ |
| SBA | LIME-AR | $0.65 \pm 0.48$ | $0.60 \pm 0.49$ | $0.53 \pm 0.23$ | $0.44 \pm 0.23$ |
| | Wachter | $\mathbf{1.00} \pm 0.00$ | $0.61 \pm 0.45$ | $\mathbf{0.30} \pm 0.24$ | $\mathbf{0.11} \pm 0.09$ |
| | CEPM | $\mathbf{1.00} \pm 0.00$ | $0.80 \pm 0.40$ | $2.24 \pm 0.01$ | $1.42 \pm 0.00$ |
| | LIME-ROAR | $0.97 \pm 0.16$ | $\mathbf{0.97} \pm 0.16$ | $4.05 \pm 0.36$ | $1.45 \pm 0.12$ |
| | LIME-DiRRAc | $0.93 \pm 0.26$ | $0.93 \pm 0.26$ | $1.10 \pm 0.11$ | $1.07 \pm 0.05$ |
| | LIME-Gaussian DiRRAc | $0.82 \pm 0.38$ | $0.80 \pm 0.38$ | $0.64 \pm 0.29$ | $0.43 \pm 0.32$ |
| Student | LIME-AR | $0.66 \pm 0.48$ | $0.53 \pm 0.45$ | $0.53 \pm 0.63$ | $0.37 \pm 0.32$ |
| | Wachter | $\mathbf{1.00} \pm 0.00$ | $0.43 \pm 0.39$ | $\mathbf{0.40} \pm 0.27$ | $\mathbf{0.20} \pm 0.14$ |
| | CEPM | $\mathbf{1.00} \pm 0.00$ | $0.70 \pm 0.46$ | $4.51 \pm 0.00$ | $2.03 \pm 0.01$ |
| | LIME-ROAR | $0.97 \pm 0.18$ | $0.95 \pm 0.20$ | $6.30 \pm 0.19$ | $1.97 \pm 0.16$ |
| | LIME-DiRRAc | $0.97 \pm 0.18$ | $\mathbf{0.97} \pm 0.18$ | $1.12 \pm 0.23$ | $1.12 \pm 0.23$ |
| | LIME-Gaussian DiRRAc | $0.69 \pm 0.46$ | $0.59 \pm 0.46$ | $0.58 \pm 0.54$ | $0.50 \pm 0.51$ |

**Concluding Remarks.** In this work, we proposed the Distributionally Robust Recourse Action (DiRRAc) framework to address the problem of recourse robustness under shifts in the parameters of the classification model. We introduced a distributionally robust optimization approach for generating a robust recourse action using a projected gradient descent algorithm. The experimental results demonstrated that our framework has the ability to generate the recourse action that has high probability of being valid under different types of data distribution shifts with a low cost. We also showed that our framework can be adapted to different model types, linear and non-linear models, and allows for actionability constraints of the recourse action.

**Remark 5.1** (Extensions). *The DiRRAc framework can be extended to hedge against the misspecification of the mixture weights $\widehat{p}$. Alternatively, the objective function of DiRRAc can be modified to minimize the worst-case component probability. These extensions are explored in Section C. Corresponding extensions for the Gaussian DiRRAc framework are presented in Section D.*

**Remark 5.2** (Choice of ambiguity set). *This paper's results rely fundamentally on the design of ambiguity sets using a Gelbrich distance on the moment space. This Gelbrich ambiguity set leads to the $\|\cdot\|_2$-regularizations of the worst-case Value-at-Risk in Lemmas 3.3 and 4.2. If we consider other moment ambiguity sets, for example, the moment bounds in Delage & Ye (2010) or the Kullback-Leibler-type sets in Taskesen et al. (2021), then these regularization equivalence are not available, and there is no trivial way to extend the results to reformulate the (Gaussian) DiRRAc framework.*

**Acknowledgments.** Viet Anh Nguyen acknowledges the generous support from the CUHK's Improvement on Competitiveness in Hiring New Faculties Funding Scheme.

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

## A  ADDITIONAL EXPERIMENT RESULTS

Here, we provide further details about the datasets, experimental settings, and additional results. Source code can be found at `https://github.com/duykhuongnguyen/DiRRAc`.

### A.1  DATASETS

**Real-world datasets.** We use three real-world datasets which are popular in the settings of robust algorithmic recourse: German credit (Dua & Graff, 2017), SBA Li et al. (2018), and Student performance Cortez & Silva (2008). We select a subset of features from each dataset:

- For the German credit dataset from the UCI repository, we choose five features: Status, Duration, Credit amount, Personal Status, and Age. We found in the descriptions of two datasets that feature Status in the data correction shift dataset corrects the coding errors in the original dataset (Dua & Graff, 2017).

- For the SBA dataset, we follow Li et al. (2018) and Upadhyay et al. (2021) and we choose 13 features: Selected, Term, NoEmp, CreateJob, RetainedJob, UrbanRural, ChgOffPrinGr, GrAppv, SBA_Appv, New, RealEstate, Portion, Recession. We use the instances during 1989-2006 as original data and the remaining instances as shifted data.

- For the Student Performance dataset, motivated by Cortez & Silva (2008), we choose G3 - final grade for deciding the label pass or fail for each student. The student who has G3 $< 12$ is labeled 0 (failed) and 1 (passed) otherwise. For input features, we choose 9 features: Age, Study time, Famsup, Higher, Internet, Health, Absences, G1, G2. We separate the dataset into the original and the geospatial shift data by 2 different schools.

We report the accuracy of the current classifiers and shifted classifiers for two types of models: logistics classifiers (LR) and MLPs classifiers (MLPs) on each dataset in Table 3.

Table 3: Accuracy of the underlying classifiers.

| Dataset | Methods | Accuracy |
|---|---|---|
| German | LR | $0.72 \pm 0.00$ |
| | MLPs | $0.76 \pm 0.01$ |
| Shifted German | LR | $0.7 \pm 0.00$ |
| | MLPs | $0.72 \pm 0.01$ |
| SBA | LR | $0.79 \pm 0.01$ |
| | MLPs | $0.93 \pm 0.02$ |
| Shifted SBA | LR | $0.77 \pm 0.01$ |
| | MLPs | $0.89 \pm 0.01$ |
| Student | LR | $0.84 \pm 0.01$ |
| | MLPs | $0.91 \pm 0.01$ |
| Shifted Student | LR | $0.91 \pm 0.00$ |
| | MLPs | $0.99 \pm 0.01$ |

**Synthetic data.** We synthesize two-dimensional data and simulate the shifted data by using $K = 3$ different shifts similar to Upadhyay et al. (2021): mean shift, covariance shift, mean and covariance shift. First, we fix the unshifted conditional distributions with $X|Y = y \sim \mathcal{N}(\mu_y, \Sigma_y) \ \forall y \in \mathcal{Y}$. For mean shift, we replace $\mu_0$ by $\mu_0^{\text{shift}} = \mu_0 + [\alpha, 0]^\top$, where $\alpha$ is a mean shift magnitude. For covariance shift, we replace $\Sigma_0$ by $\Sigma_0^{\text{shift}} = (1 + \beta)\Sigma_0$, where $\beta$ is a covariance shift magnitude. For mean and covariance shift, we replace $(\mu_0, \Sigma_0)$ by $(\mu_0^{\text{shift}}, \Sigma_0^{\text{shift}})$. We generate 500 samples for each class from the unshifted distribution with $\mu_0 = [-3; -3]$, $\mu_1 = [3; 3]$, and $\Sigma_0 = \Sigma_1 = I$.

To visualize the decision boundaries of the linear classifiers for synthetic data, we synthesize the shifted data in total 100 times including 33 mean shifts, 33 covariance shifts and 34 both shifts, then we visualize the 100 model's parameters in a two-dimensional space in Figure 4 and Figure 5.

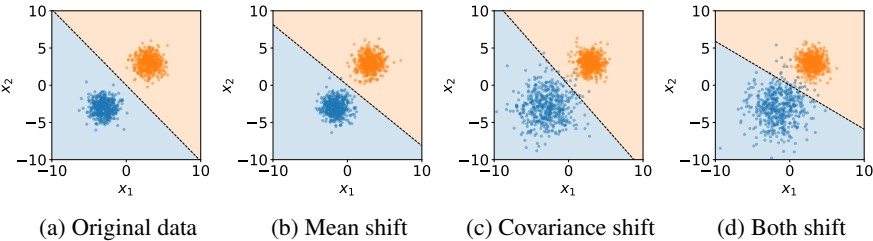

(a) Original data      (b) Mean shift      (c) Covariance shift      (d) Both shift

Figure 4: Synthetic data shifts and the corresponding model parameter shifts (decision boundaries).

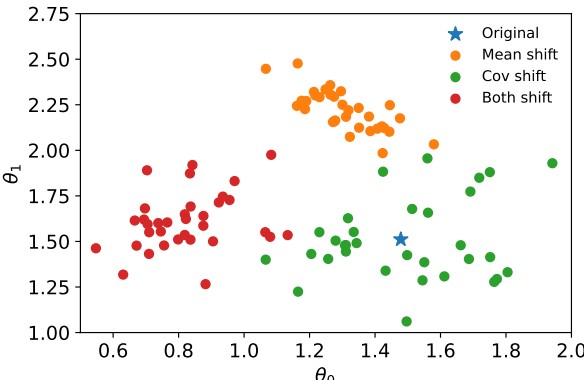

Figure 5: Parameter $\theta$ of the classifier with different types of data distribution shifts.

## A.2 EXPERIMENTAL SETTINGS

**Implementation details.** For all the baselines, we use the implementation of CARLA (Pawelczyk et al., 2021). We use the hyperparameters of AR, Wachter and CEPM that are provided by CARLA. For ROAR, we use the same parameters as in ROAR (Upadhyay et al., 2021).

**Experimental settings.** The experimental settings for the experiments in the main text are as follows:

- In Figure 3, we fix $\rho_1 = 0.1$ and vary $\delta_{\mathrm{add}} \in [0, 2.0]$ for DiRRAc. Then we fix $\delta_{\max} = 0.1$ and vary $\lambda \in [0.01, 0.2]$ for ROAR.

- In Table 1 and Table 2, we first initialize $\rho_1 = 0.1$ and we choose the $\delta_{\mathrm{add}}$ that maximizes the $M_1$ validity. We follow the same procedure as in the original paper for ROAR (Upadhyay et al., 2021): choose $\delta_{\max} = 0.1$ and find the value of $\lambda$ that maximizes the $M_1$ validity. The detailed settings are provided in Table 4.

Table 4: Parameters for the experiments with real-world data in Table 1.

| Parameters | Values |
| :---: | :---: |
| $K$ | 1 |
| $\delta_{\mathrm{add}}$ | 1.0 |
| $\widehat{p}$ | [1] |
| $\rho$ | [0.1] |
| $\lambda$ | 0.7 |
| $\zeta$ | 1 |

**Choice of number of components $K$ for real-world datasets.** To choose $K$ for real-world datasets, we use the same procedure in Section 5 to obtain 100 observations of the model parameters. Then we determine the number of components $K$ on these observations by using K-means clustering and Elbow method (Thorndike, 1953; Ketchen & Shook, 1996). Then we train a Gaussian mixture model on these observations and obtain $\widehat{p}_k, \widehat{\theta}_k, \widehat{\Sigma}_k$ for the optimal number of components $K$. The Elbow method visualization for each dataset is shown in Figure 6.

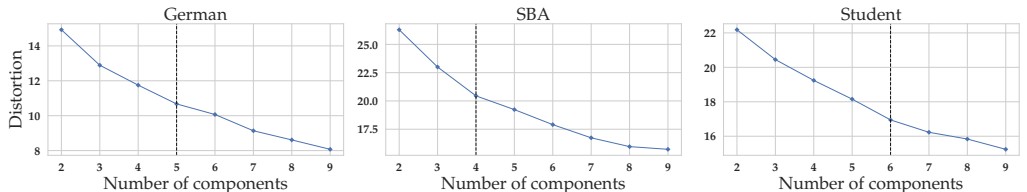

Figure 6: Elbow method for determining the optimal number of components for parameter shifts. Dashed lines represent the optimal $K$ for three real-world datasets. German Credit: Elbow at $K = 5$. SBA: Elbow at $K = 4$. Student Performace: Elbow at $K = 6$.

Table 5: Performance of DiRRAc and Gaussian DiRRAc with $K$ components on three real-world datasets.

| Dataset | Methods | $M_1$ validity | $M_2$ validity | $l_1$ cost | $l_2$ cost |
|---------|---------|----------------|----------------|------------|------------|
| German | DiRRAc ($K = 5$) | **1.00** $\pm$ 0.00 | **0.99** $\pm$ 0.07 | **1.73** $\pm$ 0.31 | 1.40 $\pm$ 0.20 |
| | Gaussian DiRRAc ($K = 5$) | **1.00** $\pm$ 0.00 | **0.99** $\pm$ 0.07 | **1.73** $\pm$ 0.31 | **1.23** $\pm$ 0.23 |
| SBA | DiRRAc ($K = 4$) | **1.00** $\pm$ 0.00 | **1.00** $\pm$ 0.00 | 1.83 $\pm$ 0.49 | 1.48 $\pm$ 0.29 |
| | Gaussian DiRRAc ($K = 4$) | **1.00** $\pm$ 0.00 | 0.99 $\pm$ 0.02 | **1.67** $\pm$ 0.68 | **0.98** $\pm$ 0.42 |
| Student | DiRRAc ($K = 6$) | **1.00** $\pm$ 0.00 | **0.96** $\pm$ 0.09 | 1.59 $\pm$ 0.33 | 1.04 $\pm$ 0.22 |
| | Gaussian DiRRAc ($K = 6$) | **1.00** $\pm$ 0.00 | 0.75 $\pm$ 0.19 | **0.82** $\pm$ 0.30 | **0.53** $\pm$ 0.21 |

The results in Table 5 indicate that as we deploy our framework with the optimal number of components $K$, then DiRRAc delivers a smaller cost in all three datasets. The $M_2$ validity of Gaussian DiRRAc slightly increases in the Student Performance dataset.

**Sensitivity analysis of hyperparameters $\delta_{\mathrm{add}}$ and $\rho_k$.** Here we analyze the sensitivity of the hyperparameters $\delta_{\mathrm{add}}$ and $\rho_k$ to the $l_1$ cost of recourses and $M_2$ validity of DiRRAc.

From the results in Figure 3, we can observe that as $\delta_{\mathrm{add}}$ increases, both the cost and the robustness of the recourse increase.

We study the sensitivity of hyperparameters $\rho_k$ to $M_2$ validity by first fixing the $\delta_{\mathrm{add}} = 0.1$ and vary $\rho_k \in [0.0, 0.5]$. According to Figure 7, we can observe that as $\rho_k$ increases, the cost of recourses rises as well, yielding in more robust recourses.

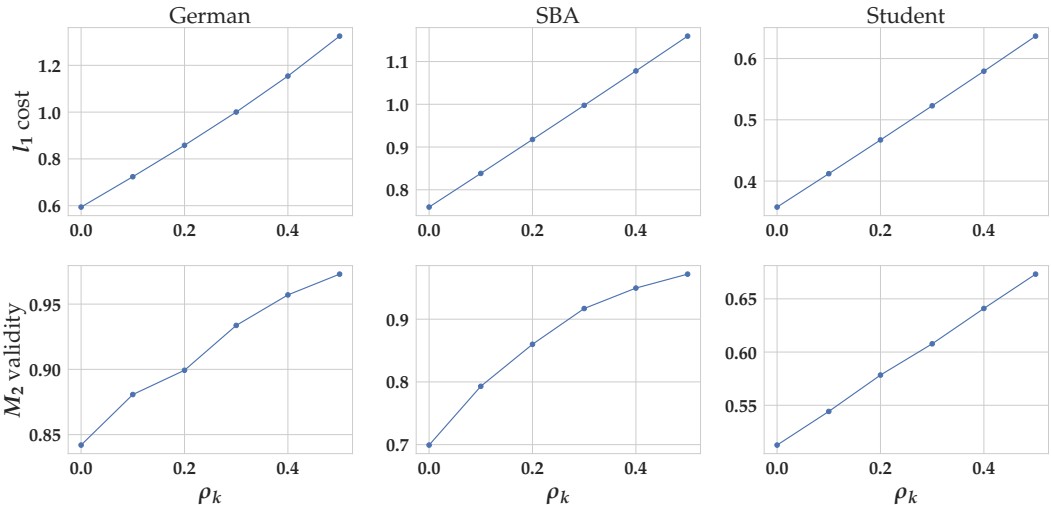

Figure 7: Sensitivity analysis of hyperparameters $\rho_k$ to $l_1$ cost and $M_2$ validity of DiRRAc.

## A.3 RESULTS ON REAL-WORLD DATA

**Experiments with prior on $\widehat{\Sigma}$.** In some cases, we presume, we may not have access to the training data. We set $\widehat{\theta}_1 = \theta_0$, where $\theta_0$ is the parameters of the original classifier. Then we choose $\widehat{\Sigma}_1 = \tau I$ with $\tau = 0.1$. We generate recourse for each input instance and compute the $M_1$ validity using the original classifier and the $M_2$ validity using the shifted classifiers. The results in Table 6 show that our methods produce the same performance while at the same time keeping the $l_1$ and $l_2$ cost lower than ROAR in all three datasets.

Table 6: Benchmark of $M_1$ validity, $M_2$ validity, $l_1$ and $l_2$ using $\widehat{\theta}_1 = \theta_0$ and $\widehat{\Sigma}_1 = 0.1I$ on different real-world datasets.

| Dataset | Methods | $M_1$ validity | $M_2$ validity | $l_1$ cost | $l_2$ cost |
|---------|---------|----------------|----------------|------------|------------|
| German | ROAR | $\mathbf{1.00} \pm 0.00$ | $0.94 \pm 0.15$ | $3.88 \pm 0.54$ | $1.61 \pm 0.22$ |
| | DiRRAc | $\mathbf{1.00} \pm 0.00$ | $0.96 \pm 0.07$ | $\mathbf{1.48} \pm 0.39$ | $\mathbf{1.34} \pm 0.41$ |
| | Gaussian DiRRAc | $\mathbf{1.00} \pm 0.00$ | $\mathbf{0.99} \pm 0.06$ | $1.58 \pm 0.29$ | $1.35 \pm 0.24$ |
| SBA | ROAR | $\mathbf{1.00} \pm 0.00$ | $\mathbf{1.00} \pm 0.00$ | $3.10 \pm 0.72$ | $1.35 \pm 0.30$ |
| | DiRRAc | $\mathbf{1.00} \pm 0.00$ | $\mathbf{1.00} \pm 0.00$ | $\mathbf{1.64} \pm 0.37$ | $1.27 \pm 0.30$ |
| | Gaussian DiRRAc | $\mathbf{1.00} \pm 0.00$ | $\mathbf{1.00} \pm 0.00$ | $\mathbf{1.64} \pm 0.37$ | $\mathbf{1.25} \pm 0.26$ |
| Student | ROAR | $\mathbf{1.00} \pm 0.00$ | $0.94 \pm 0.10$ | $2.02 \pm 0.38$ | $0.96 \pm 0.18$ |
| | DiRRAc | $\mathbf{1.00} \pm 0.00$ | $\mathbf{0.97} \pm 0.06$ | $1.81 \pm 0.19$ | $1.47 \pm 0.13$ |
| | Gaussian DiRRAc | $\mathbf{1.00} \pm 0.00$ | $0.88 \pm 0.14$ | $\mathbf{1.18} \pm 0.26$ | $\mathbf{0.82} \pm 0.18$ |

**Experiments with actionability constraints.** Using our two methods (DiRRAc and Gaussian DiRRAc) and the AR method (Ustun et al., 2019), we analyze how the actionability constraints affect the cost and validity of the recourse. We select a subset of features from each dataset and define each feature as immutable or non-decreasing as follows:

- In the German credit dataset, we select Personal status as an immutable attribute because it is challenging to impose changes in an individual's status and sex. We view age as a non-decreasing feature.

- In the SBA dataset, we select UrbanRural and Recession as two immutable attributes since it will be difficult to change these features in the near future. RetainedJob is another feature that we view as non-decreasing.

- In the Student Performance dataset, we assume that a student's Higher education would not change, and select higher education as an immutable feature. Age and Absences are considered as non-decreasing.

The above specifications are aligned with the existing numerical setup in algorithmic recourse (Ustun et al., 2019; Rawal & Lakkaraju, 2020a).

For each dataset, we run the process of generating the recourse action by adding constraints to the projected gradient descent algorithm. The experimental setup on three different real-world datasets is the same as in Section 5.

The results in Table 7 indicate that the $M_2$ validity of our 2 methods drops in the German Credit dataset. The validity in shifted data of AR also decreases in this dataset. In other datasets, the performance of our 2 methods remains the same. The $l_1$ and $l_2$ cost of DiRRAc slightly increase in the Student Performance dataset. Furthermore, there exists recourse for every input instance.

Table 7: Benchmark of $M_1$ validity, $M_2$ validity, $l_1$ and $l_2$ using actionability constraints on different real-world datasets.

| Dataset | Methods | $M_1$ validity | $M_2$ validity | $l_1$ cost | $l_2$ cost |
|---|---|---|---|---|---|
| German | AR | **1.00** $\pm$ 0.00 | 0.76 $\pm$ 0.26 | **0.61** $\pm$ 0.40 | **0.43** $\pm$ 0.25 |
| | DiRRAc | **1.00** $\pm$ 0.00 | **0.99** $\pm$ 0.06 | 1.62 $\pm$ 0.30 | 1.27 $\pm$ 0.20 |
| | Gaussian DiRRAc | **1.00** $\pm$ 0.00 | **0.99** $\pm$ 0.06 | 1.62 $\pm$ 0.30 | 1.09 $\pm$ 0.24 |
| SBA | AR | **1.00** $\pm$ 0.00 | 0.41 $\pm$ 0.18 | **0.61** $\pm$ 0.42 | **0.56** $\pm$ 0.36 |
| | DiRRAc | **1.00** $\pm$ 0.00 | **1.00** $\pm$ 0.00 | 1.74 $\pm$ 0.44 | 1.34 $\pm$ 0.40 |
| | Gaussian DiRRAc | **1.00** $\pm$ 0.00 | 0.99 $\pm$ 0.02 | 1.60 $\pm$ 0.62 | 0.98 $\pm$ 0.42 |
| Student | AR | **1.00** $\pm$ 0.00 | 0.48 $\pm$ 0.19 | **0.29** $\pm$ 0.21 | **0.26** $\pm$ 0.18 |
| | DiRRAc | **1.00** $\pm$ 0.00 | **0.95** $\pm$ 0.09 | 1.61 $\pm$ 0.31 | 1.08 $\pm$ 0.24 |
| | Gaussian DiRRAc | **1.00** $\pm$ 0.00 | 0.74 $\pm$ 0.18 | 0.81 $\pm$ 0.27 | 0.55 $\pm$ 0.21 |

**Comparison with RBR.** Here we compare our approach on the nonlinear model settings to a more recent approach on robust recourse (Nguyen et al., 2022).

Table 8: Comparison with RBR for non-linear models on real datasets.

| Dataset | Methods | $M_1$ validity | $M_2$ validity | $l_1$ cost | $l_2$ cost |
|---|---|---|---|---|---|
| German | RBR | **0.98** $\pm$ 0.13 | 0.71 $\pm$ 0.25 | **1.11** $\pm$ 0.10 | **0.50** $\pm$ 0.07 |
| | LIME-DiRRAc | 0.78 $\pm$ 0.42 | **0.75** $\pm$ 0.27 | 1.14 $\pm$ 0.27 | 1.02 $\pm$ 0.05 |
| | LIME-Gaussian DiRRAc | 0.70 $\pm$ 0.46 | 0.70 $\pm$ 0.31 | **1.11** $\pm$ 0.26 | 1.00 $\pm$ 0.06 |
| SBA | RBR | **1.00** $\pm$ 0.00 | **0.97** $\pm$ 0.12 | 1.42 $\pm$ 0.45 | 0.59 $\pm$ 0.18 |
| | LIME-DiRRAc | 0.93 $\pm$ 0.26 | 0.93 $\pm$ 0.26 | 1.10 $\pm$ 0.11 | 1.07 $\pm$ 0.05 |
| | LIME-Gaussian DiRRAc | 0.82 $\pm$ 0.38 | 0.80 $\pm$ 0.38 | **0.64** $\pm$ 0.29 | **0.43** $\pm$ 0.32 |
| Student | RBR | **1.00** $\pm$ 0.00 | 0.90 $\pm$ 0.23 | 1.02 $\pm$ 0.53 | **0.42** $\pm$ 0.20 |
| | LIME-DiRRAc | 0.97 $\pm$ 0.18 | **0.97** $\pm$ 0.18 | 1.12 $\pm$ 0.23 | 1.12 $\pm$ 0.23 |
| | LIME-Gaussian DiRRAc | 0.69 $\pm$ 0.46 | 0.59 $\pm$ 0.46 | **0.58** $\pm$ 0.54 | 0.50 $\pm$ 0.51 |

We provide the results in Table 8: we can observe that RBR has (nearly) perfect $M_1$ validity. This result is natural because RBR is designed to handle the nonlinear predictive model directly. Our methods do not have the perfect $M_1$ validity because we use the LIME approximation. However, it is important to note that in the problem of robust recourse facing future model shifts, we regard the $M_2$ validity as the most crucial metric because it is the proportion of recourse instances that are valid with respect to the shifted (future) models.

In terms of $l_1$ cost and $M_2$ validity, the results demonstrate that our method has a competitive performance compared to the existing state-of-the-art methods. In particular, LIME-DiRRAc outperforms RBR in terms of $M_2$ validity for two datasets (German and Student). In the SBA dataset, our approach has a lower $M_2$ validity, but the cost of recourses generated by our method is also lower. This result is consistent with our discussion about the $l_1$ cost and $M_2$ validity trade-off in the Appendix.

**Comparison with MINT on German Credit datasets.** We add a more recent baseline MINT proposed by Karimi et al. (2021a) for comparison purpose. MINT requires a causal graph; thus, we restrict the experiment to the German Credit dataset (the specifications of the causal graphs are not available for SBA and Student Performance). We do not consider MACE as a baseline for nonlinear

model comparison because MACE is not applicable to neural network target models due to its high computational cost. We use the same set of features as in the MINT and ROAR paper (Karimi et al., 2021a; Upadhyay et al., 2021) with four features: Sex, Age, Credit Amount and Duration. The results in Table 9 demonstrate that the recourse generated by our framework is more robust to model shifts, but it has a higher $l_1$ cost.

Table 9: Comparison with MINT on German Credit dataset.

| Methods | $M_1$ validity | $M_2$ validity | $l_1$ cost |
|---|---|---|---|
| MINT | **1.00** $\pm$ 0.00 | 0.87 $\pm$ 0.09 | **0.77** $\pm$ 0.23 |
| DiRRAc | **1.00** $\pm$ 0.00 | **0.99** $\pm$ 0.06 | 1.62 $\pm$ 0.30 |
| Gaussian DiRRAc | **1.00** $\pm$ 0.00 | **0.99** $\pm$ 0.06 | 1.62 $\pm$ 0.30 |

**Comparison with ensemble baselines.** Prior work suggested that model ensembles can be effective for out-of-distribution prediction (Ovadia et al., 2019; Fort et al., 2019). Now we explore a model ensemble method to generate recourse based on ROAR as follows. First we follow the procedure in Section 5 to obtain 100 model parameters $\theta^i$ with $i \in \{1, \ldots, 100\}$. Then we find recourse by solving the following problem:

$$x'' = \arg \min_{x'' \in \mathcal{A}} \max_{\delta \in \Delta} \max_{i \in \{1,\ldots,100\}} \ell \left( \mathcal{C}_{\theta^i_\delta} (x''), 1 \right) + \lambda c (x_0, x''),$$

where $\ell$ is the cross-entropy loss function.

Second, we use the same 100 models and generate recourse for each model independently. Then we average the ROAR recourses across those 100 models as follows.

$$x'' = \frac{1}{100} \sum_{i=1}^{100} \arg \min_{x'' \in \mathcal{A}} \max_{\delta \in \Delta} \ell \left( \mathcal{C}_{\theta^i_\delta} (x''), 1 \right) + \lambda c (x_0, x'').$$

Table 10: Benchmark of different variants of ROAR on three real-world datasets.

| Dataset | Methods | $M_1$ validity | $M_2$ validity | $l_1$ cost | $l_2$ cost |
|---|---|---|---|---|---|
| German | ROAR | **1.00** $\pm$ 0.00 | 0.94 $\pm$ 0.15 | 3.88 $\pm$ 0.54 | 1.61 $\pm$ 0.22 |
| | ROAR-Ensemble | **1.00** $\pm$ 0.00 | 0.95 $\pm$ 0.15 | 5.11 $\pm$ 0.59 | 2.12 $\pm$ 0.24 |
| | ROAR-Avg | **1.00** $\pm$ 0.00 | 0.95 $\pm$ 0.15 | 4.46 $\pm$ 0.36 | 2.00 $\pm$ 0.14 |
| | DiRRAc | **1.00** $\pm$ 0.00 | **0.99** $\pm$ 0.06 | **1.62** $\pm$ 0.30 | 1.25 $\pm$ 0.21 |
| | Gaussian DiRRAc | **1.00** $\pm$ 0.00 | **0.99** $\pm$ 0.06 | **1.62** $\pm$ 0.30 | **1.05** $\pm$ 0.23 |
| SBA | ROAR | **1.00** $\pm$ 0.00 | **1.00** $\pm$ 0.00 | 3.10 $\pm$ 0.72 | 1.35 $\pm$ 0.30 |
| | ROAR-Ensemble | **1.00** $\pm$ 0.00 | **1.00** $\pm$ 0.00 | 4.54 $\pm$ 0.95 | 1.91 $\pm$ 0.38 |
| | ROAR-Avg | **1.00** $\pm$ 0.00 | **1.00** $\pm$ 0.00 | 2.86 $\pm$ 0.70 | 1.78 $\pm$ 0.35 |
| | DiRRAc | **1.00** $\pm$ 0.00 | **1.00** $\pm$ 0.00 | 1.74 $\pm$ 0.44 | 1.34 $\pm$ 0.40 |
| | Gaussian DiRRAc | **1.00** $\pm$ 0.00 | 0.99 $\pm$ 0.02 | **1.60** $\pm$ 0.62 | **0.98** $\pm$ 0.42 |
| Student | ROAR | **1.00** $\pm$ 0.00 | 0.94 $\pm$ 0.10 | 2.02 $\pm$ 0.38 | 0.96 $\pm$ 0.18 |
| | ROAR-Ensemble | **1.00** $\pm$ 0.00 | **0.98** $\pm$ 0.05 | 3.73 $\pm$ 0.50 | 1.43 $\pm$ 0.19 |
| | ROAR-Avg | **1.00** $\pm$ 0.00 | 0.97 $\pm$ 0.10 | 2.78 $\pm$ 0.31 | 1.31 $\pm$ 0.17 |
| | DiRRAc | **1.00** $\pm$ 0.00 | 0.95 $\pm$ 0.09 | 1.55 $\pm$ 0.34 | 1.07 $\pm$ 0.23 |
| | Gaussian DiRRAc | **1.00** $\pm$ 0.00 | 0.74 $\pm$ 0.18 | **0.78** $\pm$ 0.30 | **0.54** $\pm$ 0.21 |

In Table 10, we provide results for the ROAR ensemble method as ROAR-Ensemble and the average ROAR recourses as ROAR-Avg. From this table, the $M_1$ and $M_2$ validity of ROAR-Ensemble and ROAR-Avg remain the same for all datasets. In almost every benchmark, the recourses generated by those two approaches are more costly than ROAR. In comparison with our framework, our DiRRAc and Gaussian DiRRAc methods demonstrate advantages in terms of the cost of recourses.

**More discussions about cost-validity trade-off.** Previous work about robust recourses have suggested that recourses are more robust with the expense of higher costs (Rawal et al., 2021; Upadhyay et al., 2021; Pawelczyk et al., 2020; Black et al., 2022). Our results with DiRRAc and Gaussian

DiRRAc are consistent with this suggestion. However, our framework can achieve robust and actionable recourses with a far smaller cost than ROAR (Upadhyay et al., 2021) and CEPM (Pawelczyk et al., 2020).

**Comparison of run time.** Table 11 reports the average run time: we observe that Wachter has the smallest run time, and our (Gaussian) DiRRAc has a smaller run time than ROAR in all datasets.

Table 11: Average runtime (seconds).

| Methods | German | SBA | Student |
|---|---|---|---|
| AR | 0.027 | 0.046 | 0.039 |
| Wachter | 0.006 | 0.011 | 0.006 |
| ROAR | 0.396 | 0.355 | 0.412 |
| DiRRAc | 0.208 | 0.363 | 0.244 |
| Gaussian DiRRAc | 0.091 | 0.117 | 0.124 |

### A.4 RESULTS ON SYNTHETIC DATA

We define the adaptive mean and covariance shift magnitude as $\alpha = \mu_{\text{adapt}} \times iter$, $\beta = \Sigma_{\text{adapt}} \times iter$ with $\mu_{\text{adapt}}, \Sigma_{\text{adapt}}$ are the factor of data shifts, $iter$ is the index of iterative loop of synthesizing process.

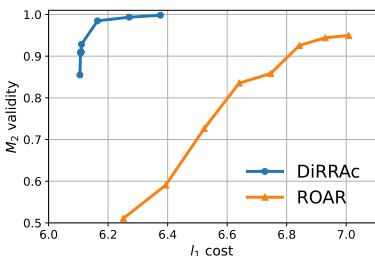

Figure 8: Comparison of $M_2$ validity as a function of the $l_1$ distance between input instance and the recourse for our DiRRAc method and ROAR on synthetic data.

For data distribution shifts, we generate mean shifts and covariance shifts 50 times each type with adaptive mean and covariance shift magnitude, with the parameters $\mu_{\text{adapt}} = \Sigma_{\text{adapt}} = 0.1$. To estimate $\widehat{\theta}_k$ and $\widehat{\Sigma}_k$, we define valid mixture weights $\widehat{p}$ and generate data for each component for 100 times with the same ratio as the mixture weight. We train 100 logistic classifiers to compute the empirical mean $\widehat{\theta}_k$ and the empirical covariance matrix $\widehat{\Sigma}_k$ for the $k$-th component. We generate a recourse for each test instance that belongs to the negative class. In Figure 8, we present the results of the cost-robustness analysis of DiRRAc and ROAR on synthetic data.

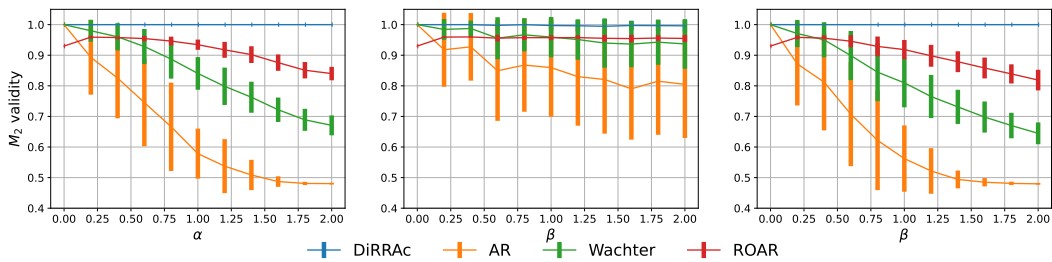

Figure 9: Impact of distribution shifts to the empirical validity. Left: mean shifts parametrized by $\alpha$; Center: covariance shifts parametrized by $\beta$; Right: Mean and covariance shifts with $\alpha = \beta$.

# B PROOFS

## B.1 PROOFS OF SECTION 3

To prove Proposition 3.4, we are using the notion of Value-at-Risk which is defined as follows.

**Definition B.1** (Value-at-Risk). *For any fixed distribution $\mathbb{Q}_k$ of $\tilde{\theta}$, the Value-at-Risk at the risk tolerance level $\beta \in (0, 1)$ of the loss $\tilde{\theta}^\top x$ is defined as*

$$\mathbb{Q}_k\text{-}\mathrm{VaR}_\beta(\tilde{\theta}^\top x) \triangleq \inf\{\tau \in \mathbb{R} : \mathbb{Q}_k(\tilde{\theta}^\top x \leq \tau) \geq 1 - \beta\}.$$

We are now ready to provide the proof of Proposition 3.4.

*Proof of Proposition 3.4.* Using the definition of the Value-at-Risk in Definition B.1, we have

$$\sup_{\mathbb{Q}_k \in \mathcal{B}_k(\widehat{\mathbb{P}}_k)} \mathbb{Q}_k(\tilde{\theta}^\top x \leq 0) = \inf\left\{\beta : \beta \in [0, 1], \sup_{\mathbb{Q}_k \in \mathcal{B}_k(\widehat{\mathbb{P}}_k)} \mathbb{Q}_k(\tilde{\theta}^\top x \leq 0) \leq \beta\right\}$$

$$= \inf\left\{\beta : \beta \in [0, 1], \sup_{\mathbb{Q}_k \in \mathcal{B}_k(\widehat{\mathbb{P}}_k)} \mathbb{Q}_k\text{-}\mathrm{VaR}_\beta(-\tilde{\theta}^\top x) \leq 0\right\}$$

By Nguyen (2019, Lemma 3.31), we can reformulate the worst-case value-at-risk as

$$\sup_{\mathbb{Q}_k \in \mathcal{B}_k(\widehat{\mathbb{P}}_k)} \mathbb{Q}_k\text{-}\mathrm{VaR}_\beta(-\tilde{\theta}^\top x) = -\widehat{\theta}_k^\top x + \sqrt{\frac{1 - \beta}{\beta}}\sqrt{x^\top \widehat{\Sigma}_k x} + \frac{\rho_k}{\sqrt{\beta}}\|x\|_2.$$

It is now easy to observe that in the first case when $-\widehat{\theta}_k^\top x + \rho_k\|x\|_2 \geq 0$, then we should have $\sup_{\mathbb{Q}_k \in \mathcal{B}_k(\widehat{\mathbb{P}}_k)} \mathbb{Q}_k(\tilde{\theta}^\top x \leq 0) = 1$.

We now consider the second case when $-\widehat{\theta}_k^\top x + \frac{\rho_k}{\sqrt{\beta}}\|x\|_2 < 0$. It is easy to see, by the monotocity of the worst-case value-at-risk with respect to $\beta$, that the minimal value $\beta^\star$ should satisfies

$$-\widehat{\theta}_k^\top x + \sqrt{\frac{1 - \beta^\star}{\beta^\star}}\sqrt{x^\top \widehat{\Sigma}_k x} + \frac{\rho_k}{\sqrt{\beta^\star}}\|x\|_2 = 0.$$

Using the transformation $t \leftarrow \sqrt{\beta^\star}$, we have

$$-\widehat{\theta}_k^\top x t + \sqrt{1 - t^2}\sqrt{x^\top \widehat{\Sigma}_k x} + \rho_k\|x\|_2 = 0.$$

By rearranging terms and then squaring up both sides, we have the equivalent quadratic equation

$$(A_k^2 + B_k^2)t^2 + 2A_k C_k t + C_k^2 - B_k^2 = 0$$

with $A_k \triangleq -\widehat{\theta}_k^\top x \leq 0$, $B_k \triangleq \sqrt{x^\top \widehat{\Sigma}_k x} \geq 0$, and $C_k \triangleq \rho_k\|x\|_2 \geq 0$ as defined in the statement of the proposition. Note, moreover, that we also have $A_k^2 \geq C_k^2$. This leads to the solution

$$t = \frac{-A_k C_k + B_k\sqrt{A_k^2 + B_k^2 - C_k^2}}{A_k^2 + B_k^2} \geq 0.$$

Thus, we find

$$f_k(x) = \left(\frac{-A_k C_k + B_k\sqrt{A_k^2 + B_k^2 - C_k^2}}{A_k^2 + B_k^2}\right)^2.$$

This completes the proof. $\qquad\square$

We now provide the proof of Theorem 3.2.

*Proof of Theorem 3.2.* We first consider the objective function $f$ of (2), which can be re-expressed as

$$
\begin{aligned}
f(x) &= \sup_{\mathbb{P} \in \mathcal{B}(\widehat{\mathbb{P}})} \mathbb{P}(\mathcal{C}_{\tilde{\theta}}(x) = 0) \\
&= \sup_{\mathbb{Q}_k \in \mathcal{B}_k(\widehat{\mathbb{P}}_k) \ \forall k} \sum_{k \in [K]} \widehat{p}_k \mathbb{Q}_k(\tilde{\theta}^\top x \le 0) \\
&= \sum_{k \in [K]} \widehat{p}_k \times \sup_{\mathbb{Q}_k \in \mathcal{B}_k(\widehat{\mathbb{P}}_k)} \mathbb{Q}_k(\tilde{\theta}^\top x \le 0) \\
&= \sum_{k \in [K]} \widehat{p}_k \times f_k(x),
\end{aligned}
$$

where the equality in the second line follows from the non-negativity of $\widehat{p}_k$, and the last equality follows from the definition of $f_k(x)$ in (5). Applying Proposition 3.4, we obtain the objective function of problem (4).

Consider now the last constraint of (2). Using the result of Proposition 3.4, this constraint is equivalent to

$$
-\widehat{\theta}_k^\top x + \rho_k \|x\|_2 < 0 \qquad \forall k \in [K].
$$

This leads to the feasible set $\mathcal{X}$ as is defined in (3). This completes the proof. $\qquad\square$

### B.2 PROOFS OF SECTION 4

To prove Theorem 4.1, we first define the following worst-case Gaussian component probability function

$$
f_k^{\mathcal{N}}(x) \triangleq \sup_{\mathbb{Q}_k \in \mathcal{B}_k^{\mathcal{N}}(\widehat{\mathbb{P}}_k)} \mathbb{Q}_k(\mathcal{C}_{\tilde{\theta}}(x) = 0) = \sup_{\mathbb{Q}_k \in \mathcal{B}_k^{\mathcal{N}}(\widehat{\mathbb{P}}_k)} \mathbb{Q}_k(\tilde{\theta}^\top x \le 0) \qquad \forall k \in [K]. \tag{11}
$$

The next proposition provides the reformulation of $f_k^{\mathcal{N}}$.

**Proposition B.2** (Worst-case probability - Gaussian). *For any $x \in \mathbb{R}^d$, any $k \in [K]$ and any $(\widehat{\theta}_k, \widehat{\Sigma}_k, \rho_k) \in \mathbb{R}^d \times \mathbb{S}_+^d \times \mathbb{R}_+$, define the following constants $A_k \triangleq -\widehat{\theta}_k^\top x$, $B_k \triangleq \sqrt{x^\top \widehat{\Sigma}_k x}$, and $C_k \triangleq \rho_k \|x\|_2$. The following holds:*

*(i) We have $f_k^{\mathcal{N}}(x) < \frac{1}{2}$ if and only if $A_k + C_k < 0$.*

*(ii) If $x$ satisfies $f_k^{\mathcal{N}}(x) < \frac{1}{2}$, then*

$$
f_k^{\mathcal{N}}(x) = 1 - \Phi\left( \frac{A_k^2 - C_k^2}{-A_k B_k + C_k \sqrt{A_k^2 + B_k^2 - C_k^2}} \right).
$$

*Proof of Proposition B.2.* We first prove Assertion (i). Pick any $\mathbb{Q}_k \in \mathcal{B}_k^{\mathcal{N}}(\widehat{\mathbb{P}}_k)$, then $\mathbb{Q}_k$ is a Gaussian distribution $\mathbb{Q}_k \sim \mathcal{N}(\theta_k, \Sigma_k)$, and thus

$$
\mathbb{Q}_k(\tilde{\theta}^\top x \le 0) = \Phi\left( \frac{-\theta_k^\top x}{\sqrt{x^\top \Sigma x}} \right).
$$

Guaranteeing $f_k^{\mathcal{N}}(x) < \frac{1}{2}$ is equivalent to guaranteeing

$$
\sup_{\mathbb{G}((\theta_k, \Sigma_k),(\widehat{\theta}_k, \widehat{\Sigma}_k)) \le \rho_k} -\theta_k^\top x \le 0.
$$

Note that we also have

$$
\sup_{\mathbb{G}((\theta_k, \Sigma_k),(\widehat{\theta}_k, \widehat{\Sigma}_k)) \le \rho_k} -\theta_k^\top x = \sup_{\theta_k : \|\theta_k - \widehat{\theta}_k\|_2 \le \rho_k} -\theta_k^\top x = -\widehat{\theta}_k^\top x + \rho_k \|x\|_2
$$

by the properties of the dual norm. This leads to the equivalent condition that $A_k + C_k < 0$.

We now prove Assertion (ii). Using the definition of the Value-at-Risk in Definition B.1, we have

$$\sup_{\mathbb{Q}_k \in \mathcal{B}_k^{\mathcal{N}}(\widehat{\mathbb{P}}_k)} \mathbb{Q}_k(\tilde{\theta}^\top x \leq 0) = \inf\left\{\beta : \beta \in [0, \frac{1}{2}), \quad \sup_{\mathbb{Q}_k \in \mathcal{B}_k^{\mathcal{N}}(\widehat{\mathbb{P}}_k)} \mathbb{Q}_k(\tilde{\theta}^\top x \leq 0) \leq \beta\right\}$$

$$= \inf\left\{\beta : \beta \in [0, \frac{1}{2}), \quad \sup_{\mathbb{Q}_k \in \mathcal{B}_k^{\mathcal{N}}(\widehat{\mathbb{P}}_k)} \mathbb{Q}_k\text{-}\mathrm{VaR}_\beta(-\tilde{\theta}^\top x) \leq 0\right\}$$

Using the result from Nguyen (2019, Lemma 3.31), we have

$$\sup_{\mathbb{Q}_k \in \mathcal{B}_k(\widehat{\mathbb{P}}_k)} \mathbb{Q}_k\text{-}\mathrm{VaR}_\beta(-\tilde{\theta}^\top x) = -\widehat{\theta}_k^\top x + t\sqrt{x^\top \widehat{\Sigma}_k x} + \rho\sqrt{1 + t^2}\|x\|_2 = A_k + B_k t + C_k\sqrt{1+t},$$

with $t = \Phi^{-1}(1-\beta)$. Taking the infimum over $\beta$ is then equivalent to finding the root of the equation

$$A_k + tB_k + C_k\sqrt{1+t^2} = 0.$$

Using a transformation $\tau = 1/t$, the above equation becomes

$$A_k\tau + B_k + C_k\sqrt{1+\tau^2} = 0$$

with solution

$$\tau = \frac{-A_kB_k + C_k\sqrt{A_k^2 + B_k^2 - C_k^2}}{A_k^2 - C_k^2} > 0.$$

Notice that $A_k + C_k < 0$, and we also have $A_k^2 > C_k^2$, thus $\tau$ is well-defined. The result now follows by noticing that $f_k^{\mathcal{N}}(x) = 1 - \Phi(t) = 1 - \Phi(1/\tau)$. $\qquad\square$

We are now ready to prove Theorem 4.1.

*Proof of Theorem 4.1.* Problem (8) is equivalent to

$$\begin{aligned} \min \quad & \sum_{k \in [K]} \widehat{p}_k \times f_k^{\mathcal{N}}(x) \\ \text{s.t.} \quad & c(x, x_0) \leq \delta \\ & f_k^{\mathcal{N}}(x) < \frac{1}{2} \qquad \forall k \in [K]. \end{aligned}$$

Applying Proposition B.2, we obtain the necessary result. $\qquad\square$

## C    EXTENSIONS OF THE DiRRAc FRAMEWORK

Throughout this section, we explore two extensions of our DiRRAc framework. In Section C.1, we study an additional layer of robustification with respect to the mixture weights $\widehat{p}$. Next, in Section C.2, we consider an alternative formulation of the objective function to minimize the worst-case component probability.

### C.1    ROBUSTIFICATION AGAINST MIXTURE WEIGHT UNCERTAINTY

The DiRRAc problem considered in Section 3 only robustifies the component distributions $\widehat{\mathbb{P}}_k$. We now discuss a plausible approach to robustify against the misspecification of the mixture weights $\widehat{p}$. Because the mixture weights should form a probability vector, it is convenient to model the perturbation in the mixture weights using the $\phi$-divergence.

**Definition C.1** ($\phi$-divergence). *Let $\phi : \mathbb{R} \to \mathbb{R}$ be a convex function on the domain $\mathbb{R}_+$, $\phi(1) = 0$, $0 \times \phi(a/0) = a \times \lim_{t\uparrow\infty} \phi(t)/t$ for $a > 0$, and $0 \times \phi(0/0) = 0$. The $\phi$-divergence $\mathbb{D}_\phi$ between two probability vectors $p$, $\widehat{p} \in \mathbb{R}_+^K$ amounts to $\mathbb{D}_\phi(p \parallel \widehat{p}) \triangleq \sum_{k \in [K]} \widehat{p}_k \times \phi(p_k/\widehat{p}_k)$.*

The family of $\phi$-divergences contains many well-known statistical divergences such as the Kullback-Leibler divergence, the Hellinger distance, etc. Further discussion on this family can be found in Pardo (2018). Distributionally robust optimization models with $\phi$-divergence ambiguity set were originally studied in decision-making problems (Ben-Tal et al., 2013; Bayraksan & Love, 2015) and

have recently gained attention thanks to their successes in machine learning tasks (Namkoong & Duchi, 2017; Hashimoto et al., 2018; Duchi et al., 2021).

Let $\varepsilon \geq 0$ be a parameter indicating the uncertainty level of the mixture weights. The uncertainty set for the mixture weights is formally defined as

$$\Delta \triangleq \left\{ p \in [0,1]^K : \mathbb{1}^\top p = 1, \ \mathbb{D}_\phi(p \parallel \widehat{p}) \leq \varepsilon \right\},$$

which contains all $K$-dimensional probability vectors which are of $\phi$-divergence at most $\varepsilon$ from the nominal weights $\widehat{p}$. The ambiguity set of the mixture distributions that hedge against the weight misspecification is

$$\mathcal{U}(\widehat{\mathbb{P}}) \triangleq \left\{ \mathbb{Q} : \ \exists p \in \Delta, \ \exists \mathbb{Q}_k \in \mathcal{B}_k(\widehat{\mathbb{P}}_k) \ \forall k \in [K] \text{ such that } \mathbb{Q} \sim (\mathbb{Q}_k, p_k) \ \right\},$$

where the component sets $\mathcal{B}_k(\widehat{\mathbb{P}}_k)$ are defined as in Section 3. The DiRRAc problem with respect to the ambiguity set $\mathcal{U}(\widehat{\mathbb{P}})$ becomes

$$
\begin{aligned}
\min \quad & \sup_{\mathbb{P} \in \mathcal{U}(\widehat{\mathbb{P}})} \mathbb{P}(\mathcal{C}_{\tilde{\theta}}(x) = 0) \\
\text{s.t.} \quad & c(x, x_0) \leq \delta \\
& \sup_{\mathbb{Q}_k \in \mathcal{B}_k(\widehat{\mathbb{P}}_k)} \mathbb{Q}_k(\mathcal{C}_{\tilde{\theta}}(x) = 0) < 1 \qquad \forall k \in [K].
\end{aligned}
\tag{12}
$$

It is important to note at this point that the feasible set of (12) coincides with the feasible set of (2). Thus, to resolve problem (12), it suffices to analyze the objective function of (12). Given the function $\phi$, we define its conjugate function $\phi^* : \mathbb{R} \to \mathbb{R} \cup \{\infty\}$ by

$$\phi^*(s) = \sup_{t \geq 0} \{ts - \phi(t)\}.$$

The next theorem asserts that the worst-case probability under $\mathcal{U}(\widehat{\mathbb{P}})$ can be computed by solving a convex program.

**Theorem C.2** (Objective value). *The feasible set of problem (12) coincides with $\mathcal{X}$. Further, for every $x \in \mathcal{X}$, the objective value of (12) equals to the optimal value of a convex optimization problem*

$$\sup_{\mathbb{P} \in \mathcal{U}(\widehat{\mathbb{P}})} \mathbb{P}(\mathcal{C}_{\tilde{\theta}}(x) = 0) = \min_{\lambda \in \mathbb{R}_+, \ \eta \in \mathbb{R}} \eta + \varepsilon \lambda + \lambda \sum_{k \in [K]} \widehat{p}_k \phi^* \left( \frac{f_k(x) - \eta}{\lambda} \right),$$

*where $f_k(x)$ are computed using Proposition 3.4.*

*Proof of Theorem C.2.* From the definition of the set $\mathcal{U}(\widehat{\mathbb{P}})$, we can rewrite $F$ using a two-layer decomposition

$$
\begin{aligned}
F(x) = \sup_{\mathbb{P} \in \mathcal{U}(\widehat{\mathbb{P}})} \mathbb{P}(\mathcal{C}_{\tilde{\theta}}(x) = 0) &= \sup_{p \in \Delta} \ \sup_{\mathbb{Q}_k \in \mathcal{B}_k(\widehat{\mathbb{P}}_k) \ \forall k} \sum_{k \in [K]} p_k \mathbb{Q}_k(\tilde{\theta}^\top x \leq 0) \\
&= \sup_{p \in \Delta} \sum_{k \in [K]} p_k \times \sup_{\mathbb{Q}_k \in \mathcal{B}_k(\widehat{\mathbb{P}}_k)} \mathbb{Q}_k(\tilde{\theta}^\top x \leq 0) \\
&= \sup_{p \in \Delta} \sum_{k \in [K]} p_k \times f_k(x),
\end{aligned}
$$

where the equality in the second line follows from the non-negativity of $p_k$, and the last equality follows from the definition of $f_k(x)$ in (5). By applying the result from Ben-Tal et al. (2013, Corollary 4.2), we have

$$
F(x) = \begin{cases}
\min \quad \eta + \varepsilon \lambda + \lambda \sum_{k \in [K]} \widehat{p}_k \phi^* \left( \dfrac{f_k(x) - \eta}{\lambda} \right) \\
\text{s.t.} \quad \lambda \in \mathbb{R}_+, \ \eta \in \mathbb{R}.
\end{cases}
$$

The proof is complete. $\qquad\square$

From the result of Theorem C.2, we can derive the gradient of the objective function of (12) using Danskin's theorem Shapiro et al. (2009, Theorem 7.21), or simply using auto-differentiation. Furthermore, $\phi^*$ is convex, and thus solving the minimization problem in Theorem C.2 can be done efficiently using convex optimization algorithms.

## C.2   Minimizing the Worst-Case Component Probability

Instead of minimizing the (total) probability of unfavorable outcome, we can consider an alternative formulation where the recourse action minimizes the worst-case *conditional* probability of unfavorable outcome over all $K$ components. Mathematically, if we opt for the component ambiguity sets $\mathcal{B}_k(\widehat{\mathbb{P}}_k)$ constructed in Section 3, then we can solve

$$
\begin{aligned}
\min \quad & \max_{k \in [K]} \sup_{\mathbb{Q}_k \in \mathcal{B}_k(\widehat{\mathbb{P}}_k)} \mathbb{Q}_k(\mathcal{C}_{\tilde{\theta}}(x) = 0) \\
\mathrm{s.\,t.} \quad & c(x, x_0) \leq \delta \\
& \sup_{\mathbb{Q}_k \in \mathcal{B}_k(\widehat{\mathbb{P}}_k)} \mathbb{Q}_k(\mathcal{C}_{\tilde{\theta}}(x) = 0) < 1 \qquad \forall k \in [K].
\end{aligned}
\tag{13a}
$$

Interestingly, problem (13a) does not involve the mixture weighs $\widehat{p}$. As a consequence, a trivial advantage of this model is that it hedges automatically against the misspecification of $\widehat{p}$. To complete, we provide its equivalent finite-dimensional form.

**Corollary C.3** (Component Probability DiRRAc). *Problem (13a) is equivalent to*

$$
\min_{x \in \mathcal{X}} \quad \max_{k \in [K]} \frac{\rho_k \widehat{\theta}_k^\top x \|x\|_2 + \sqrt{x^\top \widehat{\Sigma}_k x} \sqrt{(\widehat{\theta}_k^\top x)^2 + x^\top \widehat{\Sigma}_k x - \rho_k^2 \|x\|_2^2}}{(\widehat{\theta}_k^\top x)^2 + x^\top \widehat{\Sigma}_k x}.
\tag{13b}
$$

# D   Extensions of the Gaussian DiRRAc Framework

In this section, we leverage the results in Section C to extend the Gaussian DiRRAc framework to (i) handle the uncertainty of the mixture weight and (ii) minimize the worst-case modal probability. Remind that each individual mixture ambiguity set $\mathcal{B}_k^{\mathcal{N}}(\widehat{\mathbb{P}}_k)$ is of the form

$$
\mathcal{B}_k^{\mathcal{N}}(\widehat{\mathbb{P}}_k) = \left\{ \mathbb{Q}_k : \mathbb{Q}_k \sim \mathcal{N}(\theta_k, \Sigma_k),\ \mathbb{G}((\theta_k, \Sigma_k), (\widehat{\theta}_k, \widehat{\Sigma}_k)) \leq \rho_k \right\},
$$

which is a ball in the space of Gaussian distributions.

## D.1   Handling Mixture Weight Uncertainty - Gaussian DiRRAc

Following the notations in Section C.1, we define the set of possible mixture weights as

$$
\Delta = \left\{ p \in [0,1]^K : \mathbb{1}^\top p = 1,\ \mathbb{D}_\phi(p \| \widehat{p}) \leq \varepsilon \right\}
$$

and the ambiguity set with Gaussian information is defined as

$$
\mathcal{U}^{\mathcal{N}}(\widehat{\mathbb{P}}) = \left\{ \mathbb{Q} \ :\ \exists p \in \Delta,\ \exists \mathbb{Q}_k \in \mathcal{B}_k^{\mathcal{N}}(\widehat{\mathbb{P}}_k)\ \forall k \in [K]\ \text{such that}\ \mathbb{Q} \sim (\mathbb{Q}_k, p_k)_{k \in [K]} \right\}.
$$

The distributionally robust problem with respect to the ambiguity set $\mathcal{U}(\widehat{\mathbb{P}})$ is

$$
\begin{aligned}
\inf \quad & \sup_{\mathbb{P} \in \mathcal{U}^{\mathcal{N}}(\widehat{\mathbb{P}})} \mathbb{P}(\mathcal{C}_{\tilde{\theta}}(x) = 0) \\
\mathrm{s.\,t.} \quad & c(x, x_0) \leq \delta \\
& \sup_{\mathbb{Q}_k \in \mathcal{B}_k^{\mathcal{N}}(\widehat{\mathbb{P}}_k)} \mathbb{Q}_k(\mathcal{C}_{\tilde{\theta}}(x) = 0) < \tfrac{1}{2} \qquad \forall k \in [K].
\end{aligned}
\tag{14}
$$

Following the results in Section 4, the feasible set of (14) coincides with the set $\mathcal{X}$. It suffices now to provide the reformulation for the objective function of (14).

**Corollary D.1.** *For any $x \in \mathcal{X}$, we have*

$$
\sup_{\mathbb{P} \in \mathcal{U}^{\mathcal{N}}(\widehat{\mathbb{P}})} \mathbb{P}(\mathcal{C}_{\tilde{\theta}}(x) = 0) = \left\{
\begin{aligned}
& \inf \quad \eta + \varepsilon \lambda + \lambda \sum_{k \in [K]} \widehat{p}_k \phi^* \left( \frac{f_k^{\mathcal{N}}(x) - \eta}{\lambda} \right) \\
& \mathrm{s.\,t.} \quad \lambda \in \mathbb{R}_+,\ \eta \in \mathbb{R},
\end{aligned}
\right.
$$

*where the values $f_k^{\mathcal{N}}(x)$ are obtained in Proposition B.2.*

Corollary D.2 follows from Theorem D.2 by replacing the quantities $f_k(x)$ by $f_k^{\mathcal{N}}(x)$ to take into account the Gaussian parametric information. The proof of Corollary D.2 is omitted.

## D.2 MINIMIZING WORST-CASE COMPONENT PROBABILITY

We now consider the Gaussian DiRRAc that minimizes the worst-case modal probability of infeasibility. More concretely, we consider the recourse action obtained by solving

$$
\begin{aligned}
\inf \quad & \max_{k \in [K]} \ \sup_{\mathbb{Q}_k \in \mathcal{B}_k^{\mathcal{N}}(\widehat{\mathbb{P}}_k)} \ \mathbb{Q}_k(\mathcal{C}_{\tilde{\theta}}(x) = 0) \\
\text{s.\,t.} \quad & c(x, x_0) \leq \delta \\
& \sup_{\mathbb{Q}_k \in \mathcal{B}_k^{\mathcal{N}}(\widehat{\mathbb{P}}_k)} \ \mathbb{Q}_k(\mathcal{C}_{\tilde{\theta}}(x) = 0) < \tfrac{1}{2} \qquad \forall k \in [K].
\end{aligned}
\tag{15a}
$$

The next corollary provides the equivalent form of the above optimization problem.

**Corollary D.2.** *Problem (15a) is equivalent to*

$$
\inf_{x \in \mathcal{X}} \ \max_{k \in [K]} \left\{ 1 - \Phi\left( \frac{(\widehat{\theta}_k^\top x)^2 - \rho_k^2 \|x\|_2^2}{\widehat{\theta}_k^\top x \sqrt{x^\top \widehat{\Sigma}_k x} + \rho_k \|x\|_2 \sqrt{(\widehat{\theta}_k^\top x)^2 + x^\top \widehat{\Sigma}_k x - \rho_k^2 \|x\|_2^2}} \right) \right\}.
\tag{15b}
$$

# E PROJECTED GRADIENT DESCENT ALGORITHM

The pseudocode of the algorithm is presented in Algorithm 1. The convergence guarantee for Algorithm 1 follows from Beck (2017, Theorem 10.15), and is distilled in the next theorem.

---

**Algorithm 1** Projected gradient descent algorithm with backtracking line-search

---

**Input:** Input instance $x_0$, feasible set $\mathcal{X}_\varepsilon$ and objective function $f$
**Line search parameters:** $\lambda \in (0,1)$, $\zeta > 0$ (Default values: $\lambda = 0.7, \zeta = 1$)
**Initialization:** Set $x^0 \leftarrow \mathrm{Proj}_{\mathcal{X}_\varepsilon}(x_0)$
**for** $t = 0, \ldots, T-1$ **do**
  Find the smallest integer $i \geq 0$ such that

$$
f\left(\mathrm{Proj}_{\mathcal{X}_\varepsilon}(x^t - \lambda^i \zeta \nabla f(x^t))\right) \leq f(x^t) - \frac{1}{2\lambda^i \zeta}\|x^t - \mathrm{Proj}_{\mathcal{X}_\varepsilon}(x^t - \lambda^i \zeta \nabla f(x^t))\|_2^2.
$$

  Set $x^{t+1} = \mathrm{Proj}_{\mathcal{X}_\varepsilon}(x^t - \lambda^i \zeta \nabla f(x^t))$.
**end for**
**Output:** $x^T$

---

**Theorem E.1** (Convergence guarantee). *Let $\{x^t\}_{t=0,1,\ldots,T}$ be the sequence generated by Algorithm 1. Then, all limit points of the sequence $\{x^t\}_{t=0,1,\ldots,T}$ are stationary points of problem (4) with the modified feasible set $\mathcal{X}_\varepsilon$. Furthermore, there exists some constant $C > 0$ such that for any $T \geq 1$, we have*

$$
\min_{t=0,1,\ldots,T} \frac{\left\| x^t - \mathrm{Proj}_{\mathcal{X}_\varepsilon}\left(x^t - \zeta \nabla f(x^t)\right) \right\|_2}{\zeta} \leq \frac{C}{\sqrt{T}}.
$$

