# OpenReview forum: "Distributionally Robust Recourse Action"
_ICLR.cc/2023/Conference — ICLR 2023 poster_

### Official Review · Reviewer_7AqH · 2022-10-19

**Confidence:** 5
**Correctness:** 4
**Technical Novelty And Significance:** 3
**Empirical Novelty And Significance:** 3
**Recommendation:** 8

**Clarity, Quality, Novelty And Reproducibility:**

The paper is clearly written and of high quality. The framework is novel and well-designed. I am not doubted that the numerical experiments can not be reproduced.

**Strength And Weaknesses:**

Strength:
1. The framework is well motivated and the distributionally robust optimization approach is a suitable tool to address the issues raised.
2. The ambiguity set to capture model shifts is effective, leading the framework to a formulation that can be solved with the designed algorithm.
3. The extension to hedge against the misspecification of the mixture weights also supports that the framework is well-designed.
4. The numerical experiments are convincing.

Weakness:
My only comment is that some existing works should be cited:
1. "Distributionally robust multi-item newsvendor problems with multimodal demand distributions" addresses a similar issue of uncertain mixture weights.
2. "Sharing the value-at-risk under distributional ambiguity" also values the probability under distributional ambiguity around a reference Gaussian distribution.

**Summary Of The Paper:**

The paper proposes the Distributionally Robust Recourse Action (DiRRAc) framework that generates an optimal recourse action, which under a mixture of model shifts, enjoys the highest probability of being valid. A projected gradient descent algorithm is designed to solve the model and find an optimal recourse action.

**Summary Of The Review:**

I would recommend an acceptance of this paper: the framework is well-motivated and well-designed. The treatment is clear yet noval.

---

> ### Author Response · Authors · 2022-11-19
> **Response to Reviewer 7AqH**
>
> Dear reviewer,
>
> We would like to appreciate your efforts in reviewing our paper and providing positive feedback on our paper. Thank you for pointing out the relevant literature. We now discuss these papers in Section 2 of our manuscript, and we also highlight the difference between our setup and your suggested literature:
>
> * [1] assumed that the properties of distribution $\mathbb{P}$ are known. In particular, they assumed that $\mathbb{P}=\sum_{j=1}^m p_j \mathbb{P}_j$ is known to be a mixture of $m$ distinct distributions $\mathbb{P}_1, \ldots, \mathbb{P}_m$, with known mixture probabilities $p_1, \ldots, p_m$, and the covariance matrix $\Sigma_j$ is fixed. In our setup, we do not rely on this assumption.
>
> * [2] designed an ambiguity set around a reference Gaussian distribution while we formulate the ambiguity set for a mixture of $K$ Gaussian components to handle mixture weight uncertainty.
>
> **References**
>
> [1] Hanasusanto, Grani A., et al. "Distributionally robust multi-item newsvendor problems with multimodal demand distributions." Mathematical Programming 152.1 (2015): 1-32.
>
> [2] Chen, Zhi, and Weijun Xie. "Sharing the value-at-risk under distributional ambiguity." Mathematical Finance 31.1 (2021): 531-559.

---

### Official Review · Reviewer_UpMU · 2022-10-25

**Confidence:** 3
**Correctness:** 4
**Technical Novelty And Significance:** 3
**Empirical Novelty And Significance:** 2
**Recommendation:** 6

**Clarity, Quality, Novelty And Reproducibility:**

-The clarity and quality of the presentation and writing is very clear and easy to follow.

-The problem setting targeted in this work is interesting and somewhat novel. The proposed reformulation of the problem and methods are also somewhat novel.

-The experimental details have been included in the presentation so that reproducibility seems also possible.

**Strength And Weaknesses:**

Strength:
1. This work seems to be the first work that considers the setting of distributional robust recourse action to deal with model perturbation and changes when generating recourse actions.
2. By formulating the problem using Gelbrich distance, it proposes the reformulation of this problem in general and Gaussian framework and also the corresponding algorithms.

Weaknesses:
1. If I didn't understand wrongly, the baseline AR and Wachter has better ( smaller ) $\ell_1$ cost than the proposed DiRRAc or Gaussian DiRRAc? And for M1 validity and M2 validity, it works the same as ROAR or similar to other baselines? So the performance of the proposed methods doesn't show very convincing results with comparisons to baselines.


**Summary Of The Paper:**

This work formulates and targets the setting of distributional robust recourse action to deal with model perturbation and changes in recourse action problems. It proposes the formulation of distributional robust recourse action as a min-max problem with a Gelbrich distance as the distance metric of the uncertainty set. In addition, a corresponding projected gradient descent algorithm is proposed to solve it. Experiments have been conducted to compare with baselines using the metric cost and validity.

**Summary Of The Review:**

In summary, this is a work targeting a new setting called distributional robust recourse action to deal with model perturbation and changes, inspired by a distributionally robust optimization framework. It not only provides the formulation of this new problem but also proposes new methods DiRRAc or Gaussian-DiRRAc for it. The only concern is that the experimental results does not show a large gap between the proposed method and other baselines. So I suggest a borderline acceptance.

---

> ### Author Response · Authors · 2022-11-19
> **Response to Reviewer UpMU**
>
> Dear reviewer,
>
> We thank you for your time and efforts in reviewing and providing feedback on our paper.
>
> Regarding your concerns, we believe that our method improves significantly compared to the baselines. For instance, Table 1 and Table 2 show that our DiRRAc framework generates robust recourses with a smaller $l_{1}$ and $l_{2}$ cost than ROAR at the same level of validity in all three real-world datasets. For example, at the validity level of 0.95 for the Student dataset (Table 1), our $l_1$ cost is 23% lower than the cost of ROAR.
>
>
> Regarding the cost-validity trade-offs, Figure 3 evaluates the cost and $M_2$ validity Pareto fronts of DiRRAc and ROAR for three real-world datasets. It shows that the frontiers of DiRRAc *dominate* the frontiers of ROAR.
>
> In the comparison with AR and Wachter, our framework outperforms these two methods in terms of $M_2$ validity. It is important to note that in the problem of robust recourse facing future model shifts, we regard the $M_2$ validity as the most crucial metric because it is the proportion of recourse instances that are valid with respect to the shifted (future) models.

---

### Official Review · Reviewer_URnu · 2022-10-26

**Confidence:** 2
**Clarity, Quality, Novelty And Reproducibility:** see main review
**Correctness:** 3
**Technical Novelty And Significance:** 3
**Empirical Novelty And Significance:** 2
**Recommendation:** 6

**Strength And Weaknesses:**

Strength
1.  proposed a framework of Distributionally Robust Recourse Action (DiRRAc) for designing a recourse action that is robust to mixture shifts of the model parameters
2. Numerical experiments with both synthetic and three real-world datasets demonstrate the benefits

Weakness
1. it requires LIME for local approximation
2. Lack of comparison with new baselines



**Summary Of The Paper:**

In this paper, the authors proposed the Distributionally Robust Recourse Action (DiRRAc) framework, which generates a recourse action that has a high probability of being valid under a mixture of model shifts. The authors formulated the robustified recourse setup as a min-max optimization problem, where the max problem is specified by Gelbrich distance over an ambiguity set around the distribution of model parameters.  Numerical experiments with both synthetic and three real-world datasets demonstrate the benefits of our proposed framework over state-of-the-art recourse methods.


**Summary Of The Review:**

1 . The convergence result is just a direct application of existing result, which can hardly be counted as a contribution
2 . The analysis is based on linear case which is not really practical, which for nonlinear classifiers, it requires to use LIME to approximate, which could lead to degraded performances. .
3. The authors may want to comment on/compare with the following recent work:
Nguyen, Tuan-Duy H., et al. "Robust Bayesian Recourse." Uncertainty in Artificial Intelligence. PMLR, 2022.

---

> ### Author Response · Authors · 2022-11-19
> **Response to Reviewer URnu**
>
> Dear reviewer,
>
> Thank you for reviewing our paper and providing suggestions. Below, we address specific questions raised by the reviewer.
>
>
> > The convergence result is just a direct application of existing result, which can hardly be counted as a contribution
>
> We apologize for this confusion. In the updated draft, we remove the convergence guarantees in our contributions section.
>
> > The analysis is based on linear case which is not really practical, which for nonlinear classifiers, it requires to use LIME to approximate, which could lead to degraded performances.
>
> We agree with the reviewer that the linear approximation with LIME could lead to degraded performances. However, we believe that this is an accepted practice of recent recourse methods since LIME approximation has been employed throughout the algorithmic recourse literature [1, 2, 3]. For instance, [1] stated that their method and the associated theory (which relies on the linear model assumption) can be easily extended to nonlinear models by first generating the local linear approximation of any given nonlinear model using a linear approximation method such as LIME.
>
> > Comparison with recent work - Nguyen, Tuan-Duy H., et al. "Robust Bayesian Recourse." Uncertainty in Artificial Intelligence. PMLR, 2022. (RBR)
>
> We provide the results in the following table. We can observe that RBR has (nearly) perfect $M_1$ validity. This result is natural because RBR is designed to handle the nonlinear predictive model directly. Our methods do not have the perfect $M_1$ validity because we use the LIME approximation. However, it is important to note that in the problem of robust recourse facing future model shifts, we regard the $M_2$ validity as the most crucial metric because it is the proportion of recourse instances that are valid with respect to the shifted (future) models.
>
> In terms of $l_1$ cost and $M_2$ validity, the results demonstrate that our method has a competitive performance compared to the existing state-of-the-art method. In particular, LIME-DiRRAc outperforms RBR in terms of $M_2$ validity for two datasets (German and Student). In the SBA dataset, our approach has a lower $M_2$ validity, but the cost of recourses generated by our method is also lower. This result is consistent with our discussion about the $l_1$ cost and $M_2$ validity trade-off in Appendix A3.
>
> | Dataset  |Methods               |M1 validity                |M2 validity               |L1 cost                 |L2 cost     |
> |----------|-----------------------------------------|---------------------|---------------------|---------------------|---------------------|
> | German| RBR | 0.98 $\pm$ 0.13    | 0.71 $\pm$ 0.25    | 1.11 $\pm$ 0.10    | 0.50 $\pm$ 0.07    |
> | | LIME-DiRRAc | 0.78 $\pm$ 0.42    | 0.75 $\pm$ 0.27    | 1.14 $\pm$ 0.27    |1.02 $\pm$ 0.05    |
> || LIME-Gaussian DiRRAc | 0.70 $\pm$ 0.46    | 0.70 $\pm$ 0.31    | 1.11 $\pm$ 0.26    |1.00 $\pm$ 0.06   |
> |SBA| RBR | 1.00 $\pm$ 0.00    | 0.97 $\pm$ 0.12    | 1.42 $\pm$ 0.45    | 0.59 $\pm$ 0.18    |
> | | LIME-DiRRAc | 0.93 $\pm$ 0.26    | 0.93 $\pm$ 0.26    | 1.10 $\pm$ 0.11    |1.07 $\pm$ 0.05    |
> || LIME-Gaussian DiRRAc | 0.82 $\pm$ 0.38    | 0.80 $\pm$ 0.38    | 0.64 $\pm$ 0.29    |0.43 $\pm$ 0.32    |
> | Student| RBR | 1.00 $\pm$ 0.00    | 0.90 $\pm$ 0.23    | 1.02 $\pm$ 0.53    | 0.42 $\pm$ 0.20    |
> | | LIME-DiRRAc | 0.97 $\pm$ 0.18    | 0.97 $\pm$ 0.18    | 1.12 $\pm$ 0.23    |1.12 $\pm$ 0.23    |
> || LIME-Gaussian DiRRAc | 0.69 $\pm$ 0.46    | 0.59 $\pm$ 0.46    | 0.58 $\pm$ 0.54    |0.50 $\pm$ 0.51    |
>
> **References**
>
> [1] Ustun, Berk, Alexander Spangher, and Yang Liu. "Actionable recourse in linear classification." Proceedings of the conference on fairness, accountability, and transparency. 2019.
>
> [2] Rawal, Kaivalya, and Himabindu Lakkaraju. "Beyond individualized recourse: Interpretable and interactive summaries of actionable recourses." Advances in Neural Information Processing Systems 33 (2020): 12187-12198.
>
> [3] Upadhyay, Sohini, Shalmali Joshi, and Himabindu Lakkaraju. "Towards robust and reliable algorithmic recourse." Advances in Neural Information Processing Systems 34 (2021): 16926-16937.

---

> > ### Comment · Reviewer_URnu · 2022-12-13
> > **Thanks for the response**
> >
> > I thank the authors for the detailed response. I would like to raise my score to 6.

---

### Official Review · Reviewer_LdkR · 2022-10-29

**Confidence:** 4
**Correctness:** 4
**Technical Novelty And Significance:** 3
**Empirical Novelty And Significance:** 2
**Recommendation:** 6

**Clarity, Quality, Novelty And Reproducibility:**

The paper is written clearly and technical contributions are described thoroughly. Code is provided for reproducibility. Some implementation details are not described.

**Strength And Weaknesses:**

+ The paper deals with a practical problem; generating robust recourses is necessary for models which are to be deployed in the real world
+ Experimental results are encouraging, as DiRRAc achieves best post-shift validity for all datasets shown, and compares favorably against ROAR [1] in cost of recourses
+ Unlike ROAR [1], DiRRAc doesn’t have to specify the set of possible model parameter perturbations (\Delta), but it has its own parameters to tune/specify. I am not sure if ROAR could have achieved similar M2 validity results with different \Delta, but figure 3 seems to show DiRRAc consistently achieves lower cost recourses


- DiRRAc is not the best performing method (in M2 validity) for non linear models, which are usually the ones deployed, since it relies on the local linear approximation
- How can the initial distribution parameters be computed for classifiers which already exist?
- How is \rho_k determined? Ideally, we want to have a larger ball if we expect the model to change significantly. Is there a guideline for this?
- How costly is the projection step? A study of time taken to generate recourses vs ROAR is necessary to determine practicality

Other points
* cite the source for Definition 3.1
* possible typo on page 5, last paragraph, should it be f_k(x) <1 instead of f_k(x)\leq 0 in the reformulation?
* Citations are missing for the 3 datasets used


**Summary Of The Paper:**

The paper proposes a method to generate post-hoc recourses which remain valid under model shift, called DiRRAc. The model parameters are considered as a random vector modeled according to a mixture distribution (whose parameters are fit by taking models trained on trainset samples). Future model shifts are modeled by considering all distributions close to the initial distribution, called the ambiguity set. The paper proposes a minimax problem where inner max is over the ambiguity set and min is over possible counterfactuals. By choosing Gelbrich distance, the optimization problem is shown to be solvable, and the paper proposes a Projected GD algorithm. Experiments are conducted on 3 real-world datasets, and they compare against 4 baselines.

**Summary Of The Review:**

The paper applies distributionally robust optimization techniques to generate robust recourses in a technically sound way. Their way of quantifying model uncertainty, by using ambiguity sets, gets rid of having to specify an explicit set of possible future model parameters as in ROAR. Experimental results are convincing for linear models. DiRRAc may see limited success for highly non-linear models which can change in complicated ways.

---

> ### Author Response · Authors · 2022-11-19
> **Response to Reviewer LdkR (1/2)**
>
> Dear reviewer,
>
> Thank you for your time and efforts in reviewing our paper. We would like to clarify your concerns as follows:
>
> > DiRRAc is not the best performing method (in $M_2$ validity) for nonlinear models, which are usually the ones deployed, since it relies on the local linear approximation
>
> To compare between methods, it is recommended to examine the cost-validity trade-off, instead of looking only at the $M_2$ validity. In fact, it is easy to attain a high $M_2$ validity by simply proposing a costly recourse action.
>
> On the cost-validity trade-off: Table 2 shows that our DiRRAc recourse has a far smaller $l_{1}$ and $l_{2}$ cost than ROAR at the same level of validity in all three real-world datasets, this is the main advantage of our method compared to ROAR. In the comparison with AR, Wachter, and CEPM, our framework outperforms these three methods in terms of $M_{2}$ validity.
>
> Regarding the local linear approximation, we believe that this is an accepted practice of recent recourse methods because LIME approximation has been employed throughout the algorithmic recourse literature [1, 2, 3]. For instance, [1] stated that their method and the associated theory (which relies on the linear model assumption) can be easily extended to nonlinear models by first generating the local linear approximation of any given nonlinear model using a linear approximation method such as LIME.
>
> > How can the initial distribution parameters be computed for classifiers which already exist?
>
> For the classifiers which already exist, we first use LIME to generate the linear approximation parameterized by $\theta_0$.
>
> In the case that we have no access to the training data, we set $\hat \theta_1 = \theta_0$ and $ \hat \Sigma_1 = \tau I$. We already conducted an experiment in this case and show the results in Appendix A (Table 6).
>
> In the case that training data are available, we compute the $\hat \theta_1$ and $\hat \Sigma_1$ with the same procedure in Section 5 of the main paper: We split 80\% of the original dataset and train a logistic classifier. This process is repeated independently 100 times to obtain 100 observations of the model parameters. Then we compute the empirical mean and covariance matrix for $(\hat \theta_1, \hat \Sigma_1)$.
>
>
> > How is $\rho_k$ determined?
> There are many criteria and tuning procedures that we can employ to choose $\rho_k$.
>
> One possible criterion is to choose $\rho_k$ to satisfy certain theoretical performance guarantees. For example, we can choose $\rho_k$ so that the ambiguity set contains the true distribution of the (future) model parameters with high probability. Tuning $\rho_k$ in this direction can be done by leveraging existing bounds from [4] for the Gelbrich distance.
>
> For practical purposes, and if data is available, we can resort to cross-validation methods to tune $\rho_k$ in order to maximize the out-of-sample performance [5]. Running cross-validation, unfortunately, can be computationally intensive.
>
> Apparently, in our paper, we do not determine $\rho_k$ using the abovementioned directions. Instead, we decide to run our methods over different values of $\rho_k$ in order to study the cost-validity trade-off. Analyzing the cost-validity trade-off is more informative to the users, especially in consequential domains of recourses.

---

> > ### Author Response · Authors · 2022-11-19
> > **Response to Reviewer LdkR (2/2)**
> >
> > > How costly is the projection step? A study of time taken to generate recourses vs ROAR is necessary to determine the practicality
> >
> > The solvers such as Mosek and GUROBI employ a primal-dual interior-point algorithm to solve the SOCP. This problem can be solved to $\epsilon$ accuracy in $\mathcal{O}(d^{3.5} \log(\epsilon^{-1}))$ where $d$ is the dimension of the input instance.
> >
> > The following table reports the average run time (seconds/sample) of our two methods and ROAR: we observe that our DiRRAc has a smaller run time than ROAR in the German and Student datasets, and our Gaussian DiRRAc outperforms ROAR in terms of run time in all datasets.
> >
> >
> > | Methods  |German               |SBA               |Student               |
> > |----------|-----------------------------------------|---------------------|---------------------|
> > | ROAR| 0.396 | 0.355    | 0.412    |
> > | DiRRAc| 0.208 | 0.363    | 0.244    |
> > | Gaussian DiRRAc| 0.091| 0.117| 0.124    |
> >
> > > Possible typo on page 5, last paragraph, should it be $f_k(x) <1$ instead of $f_k(x)\leq 0$ in the reformulation?
> >
> > Thanks for your comment. We have fixed the typo in our revision.
> >
> > > Citations for Definition 3.1 and 3 datasets used
> >
> > We have added the citations for Definition 3.1 and 3 datasets in our revision.
> >
> > **References**
> >
> > [1] Ustun, Berk, Alexander Spangher, and Yang Liu. "Actionable recourse in linear classification." Proceedings of the conference on fairness, accountability, and transparency. 2019.
> >
> > [2] Rawal, Kaivalya, and Himabindu Lakkaraju. "Beyond individualized recourse: Interpretable and interactive summaries of actionable recourses." Advances in Neural Information Processing Systems 33 (2020): 12187-12198.
> >
> > [3] Upadhyay, Sohini, Shalmali Joshi, and Himabindu Lakkaraju. "Towards robust and reliable algorithmic recourse." Advances in Neural Information Processing Systems 34 (2021): 16926-16937.
> >
> > [4] Nguyen, Viet Anh, et al. "Mean-covariance robust risk measurement." arXiv preprint arXiv:2112.09959 (2021).
> >
> > [5] Mohajerin Esfahani, Peyman, and Daniel Kuhn. "Data-driven distributionally robust optimization using the Wasserstein metric: Performance guarantees and tractable reformulations." Mathematical Programming 171.1 (2018): 115-166.

---

### Author Response · Authors · 2022-12-01
**Rebuttal reminder**

Dear Reviewers and AC,

We would like to thank you again for your time and effort in reviewing our paper. We sincerely hope you have had a chance to read our responses, and we look forward to receiving your feedback about whether we have addressed your concerns regarding our rebuttals. We are happy to discuss if you still have any concerns.

Authors

---

### Decision · Program_Chairs · 2023-01-20

**Decision:**

Accept: poster

**Justification For Why Not Higher Score:**


The method relies on the distribution assumption of the parameters as mixture of Gaussian, which might be limited to practice. This should be justified and discussed in details.

**Justification For Why Not Lower Score:**

All the reviewers agree the paper is novel for recourse action problem and the empirical performance is promising.


**Metareview: Summary, Strengths And Weaknesses:**


In this paper, the authors consider the recourse action problem with the uncertainty of the model, and provide a solution based on the distributional robust optimization framework. The authors then extend the method for nonlinear models with local interpretable linear model. The method is justified empirically on several datasets.

All the reviewers agree the paper is novel for recourse action problem and the empirical performance is promising.

It will be better if the authors can discuss the mixture of Gaussian distribution assumption for the parameters, and the practical setup in the optimization.

**Note From Pc:**

if the above contains the word "oral" or "spotlight" please see: "oral" presentation means -> notable-top-5% and "spotlight" means -> notable-top-25%. As stated in our emails, we are disassociating presentation type from AC recommendations